# The genomic landscape of metastatic castration-resistant prostate cancers reveals multiple distinct genotypes with potential clinical impact

Lisanne F. van Dessel[1,21], Job van Riet[1,2,3,21], Minke Smits[4], Yanyun Zhu[5,6], Paul Hamberg[7], Michiel S. van der Heijden[8,9,10], Andries M. Bergman[5,10], Inge M. van Oort[11], Ronald de Wit[1], Emile E. Voest[6,8,10], Neeltje Steeghs[8,10], Takafumi N. Yamaguchi[12], Julie Livingstone[12], Paul C. Boutros[12,13,14,15,16,17], John W.M. Martens[1,8], Stefan Sleijfer[1,8], Edwin Cuppen[18,19], Wilbert Zwart[5,6,20], Harmen J.G. van de Werken[2,3], Niven Mehra[4,22] & Martijn P. Lolkema[1,8,22]*

Metastatic castration-resistant prostate cancer (mCRPC) has a highly complex genomic landscape. With the recent development of novel treatments, accurate stratification strategies are needed. Here we present the whole-genome sequencing (WGS) analysis of fresh-frozen metastatic biopsies from 197 mCRPC patients. Using unsupervised clustering based on genomic features, we define eight distinct genomic clusters. We observe potentially clinically relevant genotypes, including microsatellite instability (MSI), homologous recombination deficiency (HRD) enriched with genomic deletions and *BRCA2* aberrations, a tandem duplication genotype associated with $CDK12^{-/-}$ and a chromothripsis-enriched subgroup. Our data suggests that stratification on WGS characteristics may improve identification of MSI, $CDK12^{-/-}$ and HRD patients. From WGS and ChIP-seq data, we show the potential relevance of recurrent alterations in non-coding regions identified with WGS and highlight the central role of AR signaling in tumor progression. These data underline the potential value of using WGS to accurately stratify mCRPC patients into clinically actionable subgroups.

[1] Department of Medical Oncology, Erasmus MC Cancer Institute, Erasmus University Medical Center Rotterdam, Rotterdam, The Netherlands. [2] Cancer Computational Biology Center, Erasmus MC Cancer Institute, Erasmus University Medical Center Rotterdam, Rotterdam, The Netherlands. [3] Department of Urology, Erasmus MC Cancer Institute, Erasmus University Medical Center Rotterdam, Rotterdam, The Netherlands. [4] Department of Medical Oncology, Radboud University Nijmegen Medical Center, Nijmegen, The Netherlands. [5] Division on Oncogenomics, The Netherlands Cancer Institute, Amsterdam, The Netherlands. [6] Oncode Institute, Utrecht, The Netherlands. [7] Department of Internal Medicine, Franciscus Gasthuis & Vlietland, Rotterdam, The Netherlands. [8] Center for Personalized Cancer Treatment, Rotterdam, The Netherlands. [9] Division of Molecular Carcinogenesis, The Netherlands Cancer Institute, Amsterdam, The Netherlands. [10] Department of Medical Oncology, The Netherlands Cancer Institute, Amsterdam, The Netherlands. [11] Department of Urology, Radboud University Nijmegen Medical Center, Nijmegen, The Netherlands. [12] Informatics and Biocomputing Program, Ontario Institute for Cancer Research, Toronto, Canada. [13] Department of Medical Biophysics, University of Toronto, Toronto, Canada. [14] Department of Pharmacology and Toxicology, University of Toronto, Toronto, Canada. [15] Department of Human Genetics, University of California Los Angeles, Los Angeles, USA. [16] Department of Urology, University of California Los Angeles, Los Angeles, USA. [17] Jonsson Comprehensive Cancer Centre, University of California Los Angeles, Los Angeles, USA. [18] Center for Molecular Medicine and Oncode Institute, University Medical Center Utrecht, Utrecht, The Netherlands. [19] Hartwig Medical Foundation, Amsterdam, The Netherlands. [20] Laboratory of Chemical Biology and Institute for Complex Molecular Systems, Department of Biomedical Engineering, Eindhoven University of Technology, Eindhoven, The Netherlands. [21] These authors contributed equally: Lisanne F. van Dessel, Job van Riet. [22] These authors jointly supervised this work: Niven Mehra, Martijn P. Lolkema. *email: m.lolkema@erasmusmc.nl

Prostate cancer is known to be a notoriously heterogeneous disease and the genetic basis for this interpatient heterogeneity is poorly understood[1,2]. The ongoing development of new therapies for metastatic prostate cancer that target molecularly defined subgroups further increases the need for accurate patient classification and stratification[3–5]. Analysis of whole-exome sequencing data of metastatic prostate cancer tumors revealed that 65% of patients had actionable targets in non-androgen receptor related pathways, including PI3K, Wnt, and DNA repair[6]. Several targeted agents involved in these pathways, including mTOR/AKT pathway inhibitors[7] and PARP inhibitors[8], are currently in various phases of development and the first clinical trials show promising results. Therefore, patients with metastatic prostate cancer could benefit from better stratification to select the most appropriate therapeutic option. More extensive analysis using whole-genome sequencing (WGS)-based classification of tumors may be useful to improve selection of patients for different targeted therapies. The comprehensive nature of WGS has many advantages, including the detection of mutational patterns, as proven by the successful treatment of patients with high-tumor mutational burden with immune checkpoint blockade therapy[9–12]. Moreover, WGS unlike exome sequencing, can detect structural variants and aberrations in noncoding regions, both important features of prostate cancer.

The stratification of prostate cancer patients, based on differences in the mutational landscape of their tumors, has mainly focused on mutually exclusive mutations, copy-number alterations, or distinct patterns in RNA-sequencing caused by the abundant TMPRSS2-ERG fusion, which is recurrent in 50% of primary prostate tumors[6,13–18]. More recently, WGS of metastatic prostate cancer tumors demonstrated that structural variants arise from specific alterations such as $CDK12^{-/-}$ and $BRCA2^{-/-}$ genotypes, and are strongly associated with genome-wide events such as large tandem duplications or small genomic deletions, respectively[19–23]. Advances in WGS analysis and interpretation have revealed rearrangement signatures in breast cancer relating to disease stage, homologous recombination deficiency (HRD), and BRCA1/BRCA2 defects based on size and type of structural variant[22,24]. Thus, WGS enables the identification of patterns of DNA aberrations (i.e., genomic scars) that may profoundly improve classification of tumors that share a common etiology, if performed in a sufficiently powered dataset.

In this study, we analyzed the WGS data obtained from 197 metastatic castration-resistant prostate cancer (mCRPC) patients. We describe the complete genomic landscape of mCRPC, including tumor specific single- and multi-nucleotide variants (SNVs and MNVs), small insertions and deletions (InDels), copy-number alterations (CNAs), mutational signatures, kataegis, chromothripsis, and structural variants (SVs). Next, we compared the mutational frequency of the detected driver genes and genomic subgroups with an unmatched WGS cohort of primary prostate cancer ($n = 210$), consisting of exclusively of Gleason score 6–7 tumors[15,25]. We investigated the presence of possible driver genes by analyzing genes with enriched (non-synonymous) mutational burdens and recurrent or high-level copy-number alterations[26,27]. By utilizing various basic genomic features reflecting genomic instability and employing unsupervised clustering, we were able to define eight distinct genomic subgroups of mCRPC patients. We combined our genomic findings with AR, FOXA1, and H3K27me ChIP-seq data, and confirmed that important regulators of AR-mediated signaling are located in non-coding regions with open chromatin and highlight the central role of AR signaling in tumor progression.

## Results

### Characteristics of the mCRPC cohort and sequencing approach. 
We analyzed fresh-frozen metastatic tumor samples and matched blood samples from 197 castration-resistant prostate cancer patients using WGS generating to date the largest WGS dataset for mCRPC (Fig. 1a). Clinical details on biopsy site, age, and previous treatments of the included patients are described in Fig. 1b, c and Supplementary Table 2. WGS data was sequenced to a mean coverage of 104X in tumor tissues and 38X in peripheral blood (Supplementary Fig. 1a). The median estimated tumor cell purity using in silico analysis of our WGS data was 62% (range: 16–96%; Supplementary Fig. 1b). Tumor cell purity correlated weakly with the frequency of called SNVs (Spearman correlation; rho = 0.2; $p = 0.005$), InDels (Spearman correlation; rho = 0.35; $p < 0.001$), MNVs (Spearman correlation; rho = 0.25; $p < 0.001$) and structural variants (Spearman correlation; rho = 0.22; $p = 0.002$; Supplementary Fig. 1c).

### Landscape of mutational and structural variants in mCRPC.
The median tumor mutational burden (TMB) at the genomic level (SNVs and InDels per Mbp) was 2.7 in our mCRPC cohort, including 14 patients with high TMB (>10). We found a median of 6621 SNVs (IQR: 5048–9109), 1008 small InDels (IQR: 739–1364), 55 MNVs (IQR: 34–86) and 224 SVs (IQR: 149–370) per patient (Supplementary Fig. 2a–c). We observed a highly complex genomic landscape consisting of multiple driver mutations and structural variants in our cohort.

We confirmed that known driver genes of prostate cancer were enriched for non-synonymous mutations (Fig. 2 and Supplementary Fig. 2e)[13,15,28]. In total, we detected 11 genes enriched with non-synonymous mutations: TP53, AR, FOXA1, SPOP, PTEN, ZMYM3, CDK12, ZFP36L2, PIK3CA, and APC. ATM was mutated in 11 samples, but after multiple-testing correction appeared not to be enriched.

Our copy-number analysis revealed distinct amplified genomic regions, including 8q and Xq and deleted regions including 8p, 10q, 13q, and 17p (Supplementary Fig. 2d). Well-known prostate cancer driver genes[8,16], such as AR, PTEN, TP53, and RB1, are located in these regions. In addition to large-scale chromosomal copy-number alterations, we could identify narrow genomic regions with recurrent copy-number alterations across samples, which could reveal important prostate cancer driver genes (Supplementary data file 1).

TMPRSS2-ERG gene fusions were the most common fusions in our cohort ($n = 84$ out of 197; 42.6%) and were the majority of ETS family fusions ($n = 84$ out of 95; 88.4%; Fig. 2 and Supplementary Fig. 3). This is comparable to primary prostate cancer, where ETS fusions are found in approximately 50% of tumors[13,15]. The predominant break point was located upstream of the second exon of ERG, which preserves its ETS-domain in the resulting fusion gene.

In 42 patients (21.3%), we observed regional hypermutation (kataegis; Fig. 2 and Supplementary Fig. 4). In addition, we did not observe novel mutational signatures specific for metastatic disease or possible pre-treatment histories (Supplementary Fig. 5)[29].

To further investigate whether our description of the genome-wide mutational burden and observed alterations in drivers and/or subtype-specific genes in mCRPC were metastatic specific, we compared our data against an unmatched WGS cohort of primary prostate cancer ($n = 210$)[15,25], consisting of Gleason score 6–7 disease. Comparison of the median genome-wide TMB (SNVs and InDels per Mbp) revealed that the TMB was roughly 3.8 times higher in mCRPC (Fig. 3a) and the frequency of structural variants was also higher between disease stages (Fig. 3b), increasing as disease progresses. Analysis on selected driver and subtype-specific genes showed that the mutational frequency of several genes (AR, TP53, MYC, ZMYM3, PTEN, PTPRD, ZFP36L2, ADAM15, MARCOD2, BRIP1, APC, KMT2C, CCAR2, NKX3-1, C8orf58, and

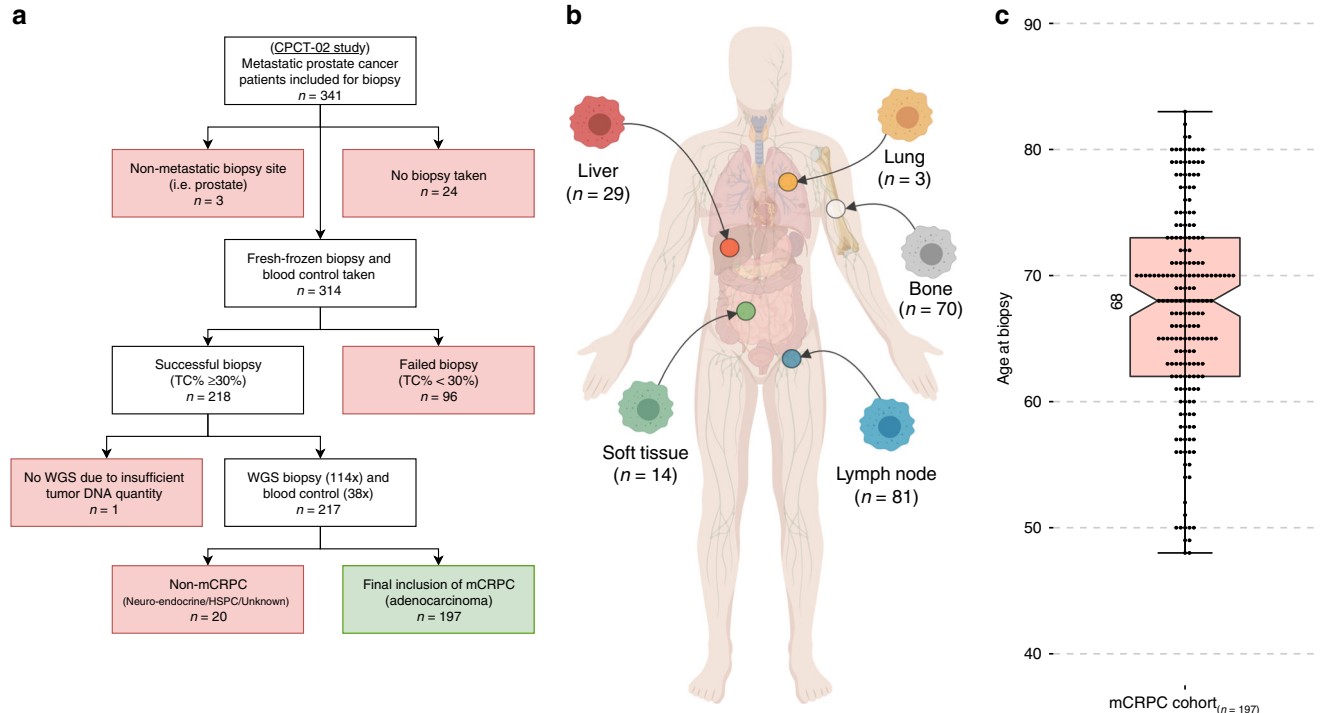

**Fig. 1** Overview of study design and patient cohort ($n = 197$). **a** Flowchart of patient inclusion. From the CPCT-02 cohort, patients with metastatic prostate cancer were selected. Patients were excluded if data from metastatic samples were not available and if clinical data indicated that patients had hormone-sensitive or neuro-endocrine prostate cancer or unknown disease status at the time of analysis. **b** Overview of the biopsy sites. Number of biopsies per metastatic site analyzed with WGS. **c** Age of patients at biopsy. Bee-swarm boxplot with notch of the patient age distribution. Boxplot depicts the upper and lower quartiles, with the median shown as a solid line; whiskers indicate 1.5 times the interquartile range (IQR). Data points outside the IQR are shown

*RYBP*) was significantly altered ($q \leq 0.05$) between the primary and metastatic cohorts (Fig. 3c–e). All genes for which we observed significant differences in mutational frequency, based on coding mutations, were enriched in mCRPC (Fig. 3d). We did not identify genomic features that were specific for the metastatic setting, beyond androgen deprivation therapy-specific aberrations revolving *AR* (no aberrations in hormone-sensitive setting versus 137 aberrations in castration-resistant setting). We cannot exclude from these data that matched sample analysis or larger scale analysis could reveal such aberrations.

We next determined whether previous treatments affect the mutational landscape. Using treatment history information, we grouped prior secondary anti-hormonal therapy, taxane-based chemotherapy and systemic radionucleotide therapy into different groups (Supplementary Fig. 6). This analysis did not reveal systematic biases due to pre-treatment in aberrations, such as TMB, kataegis, chromothripsis, *ETS* fusions, or somatically altered genes (Supplementary data file 1).

**The role of the AR-pathway in mCRPC.** Focusing on the AR-pathway revealed that aberrant AR signaling occurred in 80% of our patients. In 57.3% of patients both *AR* and the *AR*-enhancer (~66.13 Mb on chromosome X; located about 629 kbp upstream of the *AR* gene[20]) were affected (Fig. 4a). In an additional 6.6% and 14.7% of tumors only *AR* gene alterations or *AR*-enhancer amplification occurred, respectively. The percentage of mCRPC patients with the exclusive *AR*-enhancer amplification (29 out of 197; 14.7%) versus exclusively *AR*-locus amplification (13 out of 197; 6.6%) is similar to previous observations, which showed 21 out of 94 CRPC patients (10.3%) with exclusively *AR*-enhancer amplification versus 4 out of 94 CRPC patients (4.3%) with exclusively AR-locus amplification[20]. Concurrent amplification of

the *AR* gene and the *AR*-enhancer was not necessarily of equal magnitude, which resulted in differences in copy number enrichment of these loci (Fig. 4b).

To date, no AR ChIP-seq data has been reported in human mCRPC samples and evidence of increased functional activity of the amplified enhancer thus far is based on cell line models[30]. To resolve this, we performed AR ChIP-seq on two selected mCRPC patient samples with *AR*-enhancer amplification based on WGS data. As controls we used two prostate cancer cell-lines (LNCaP and VCaP) and three independent primary prostate cancer samples that did not harbor copy-number alterations at this locus (Supplementary Fig. 7)[31]. We observed active enhancer regions (H3K27ac) in the castration-resistant setting, co-occupied by AR and FOXA1, at the amplified *AR*-enhancer. This is substantially stronger when compared to the hormone-sensitive primary prostate cancer samples without somatic amplifications (Fig. 4c and Supplementary Fig. 7). Furthermore, a recurrent focal amplification in a non-coding region was observed at 8q24.21 near *PCAT1*. This locus bears similar epigenetic characteristics to the *AR*-enhancer with regard to H3K27ac and, to a lesser extent, binding of AR and/or FOXA1 in the mCRPC setting (Fig. 4d and Supplementary Fig. 7).

**WGS-based stratification defines genomic subgroups in mCRPC.** Our comprehensive WGS data and large sample size enabled us to perform unsupervised clustering on several WGS characteristics to identify genomic scars that can define subgroups of mCRPC patients. We clustered our genomic data using the total number of SVs, relative frequency of SV category (translocations, inversions, insertions, tandem duplications, and deletions), genome-wide TMB encompassing SNV, InDels and MNV, and tumor ploidy. Prior to clustering, we subdivided tandem duplications and deletions into two major categories based on the respective

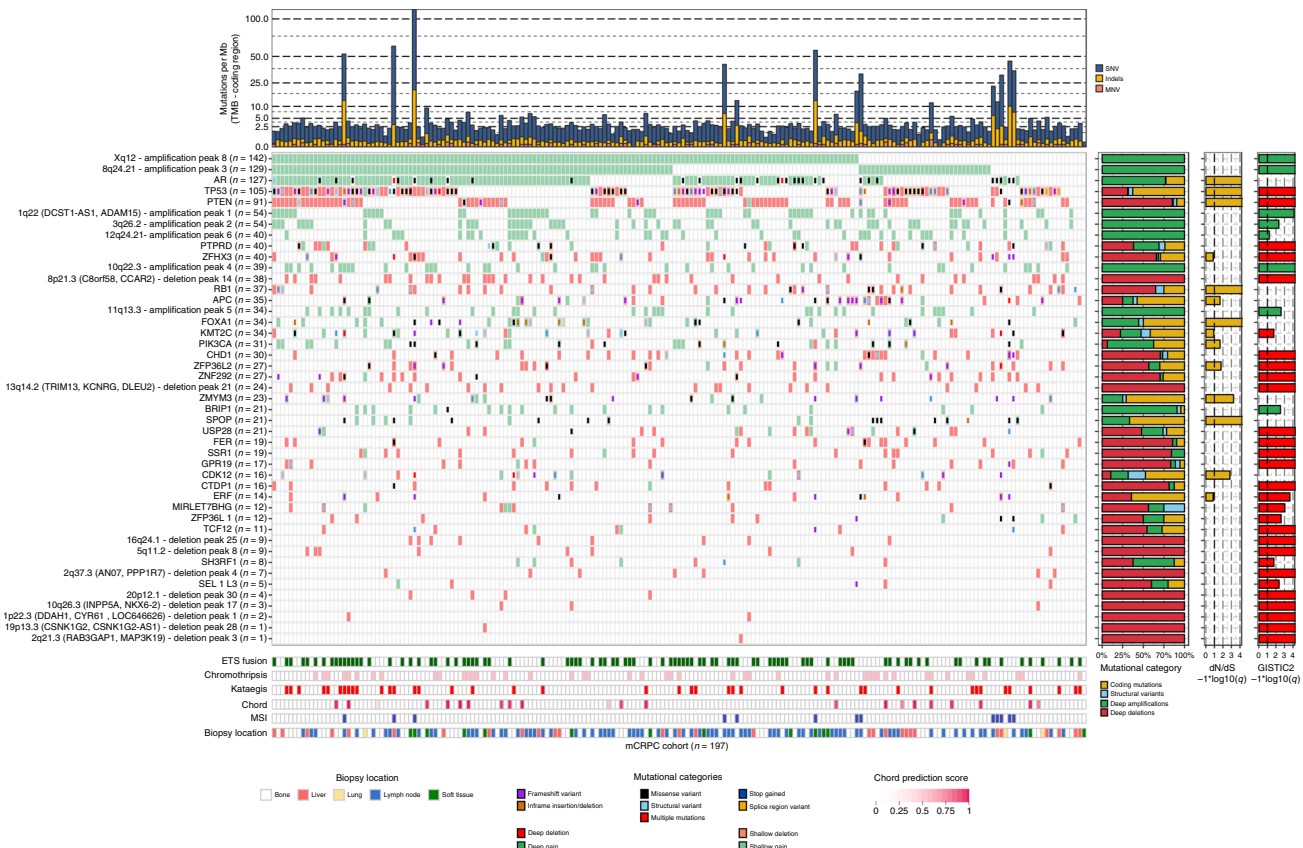

**Fig. 2** mCRPC shows multiple recurrent somatic alterations affecting several oncogenic pathways. Based on dN/dS ($q \leq 0.1$) and GISTIC2 focal peak ($q \leq 0.1$) criteria, we show the genes and focal genomic foci that are recurrently mutated, amplified, or deleted in our mCRPC cohort of 197 patients. The upper track (top bar plot) displays the number of genomic mutations per Mbp (TMB) per SNV (blue), InDel (yellow), and MNV (orange) category in coding regions (square-root scale). Samples are sorted based on mutual-exclusivity of the depicted genes and foci. The heatmap displays the type of mutation(s) per sample; (light-)green or (light-)red backgrounds depict copy-number aberrations while the inner square depicts the type of (coding) mutation(s). Relative proportions of mutational categories (coding mutations [SNV, InDels, and MNV] (yellow), SV (blue), deep amplifications [high-level amplifications resulting in many additional copies] (green), and deep deletions [high-level losses resulting in (near) homozygous losses] (red)) per gene and foci are shown in the bar plot next to the heatmap. Narrow GISTIC2 peaks covering $\leq 3$ genes were reduced to gene-level rows if one of these genes is present in the dN/dS ($q \leq 0.1$) analysis or is a known oncogene or tumor-suppressor. For GISTIC2 peaks covering multiple genes, only deep amplifications and deep deletions are shown. Recurrent aberrant focal genomic foci in gene deserts are annotated with their nearest gene. Significance scores ($-1{*}\log10$ ($q$)) of the dN/dS and GISTIC2 analysis are shown on the outer-right bar plots; bars in the GISTIC2 significance plot are colored red if these foci were detected as a recurrent focal deletion and green if detected as a recurrent focal gain. Per sample, the presence of (predicted) ETS fusions (green), chromothripsis (light pink), kataegis (red), CHORD prediction score (HR-deficiency) (pink gradient), MSI status (dark blue), and biopsy location are shown as bottom tracks

genomic size of the aberration (smaller and larger than 100 kbp) since previous studies revealed distinctions based on similar thresholds for these structural variants in relation to specific-mutated genes[19–21,32]. Similarly, we observed a difference in genomic size and number in our subgroups of mCRPC patients (Supplementary Fig. 8).

This analysis defined eight distinct subgroups (Figs. 5, 6 and Supplementary Figs. 8–11): (A) microsatellite instability (MSI) signature with high TMB and association with mismatch repair deficiency; (B) tandem duplication (>100 kbp) phenotype associated with biallelic *CDK12* inactivation; (D) homologous recombination deficiency (HRD) features with many deletions (>100 kbp) and association with (somatic) mutations in BRCAness-associated genes; this was supported by high HR-deficiency scores (CHORD; Supplementary Figs. 8 and 9); (F) chromothripsis; C, E, G, H); non-significant genomic signature without any currently known biological association. Table 1 summarizes the key features of each subgroup.

Clusters A and B represent previously identified genomic subgroups (MSI and $CDK12^{-/-}$)[6,19,21,33]. In cluster B, only two patients were allocated to this subgroup without a specific somatic mutation in the identifying gene. The well-known mismatch repair genes: *MLH1*, *MSH2*, and *MSH6* are among the cluster-specific-mutated genes in cluster A (Fig. 6a). Twelve out of these thirteen patients had at least one inactivating alteration in one of these genes (Fig. 6b). Interestingly, cluster B ($CDK12^{-/-}$) harbors two patients without non-synonymous *CDK12* mutation or copy-number alteration; the cause of their tandem duplication phenotype is currently unknown (Fig. 6b). Cluster D shows significant features of HRD, specifically biallelic *BRCA2* inactivation (Supplementary Fig. 12), mainly mutational signature 3, enrichment of deletions (<100 kbp) and is supported by high HR-deficiency scores (CHORD) (Supplementary Figs. 8 and 9)[22,34]. Remarkably, seven out of twenty-two patients did not have a biallelic *BRCA2* inactivation. However, four of these patients showed at least one (deleterious) aberration in other BRCAness-related genes (Fig. 6b)[35].

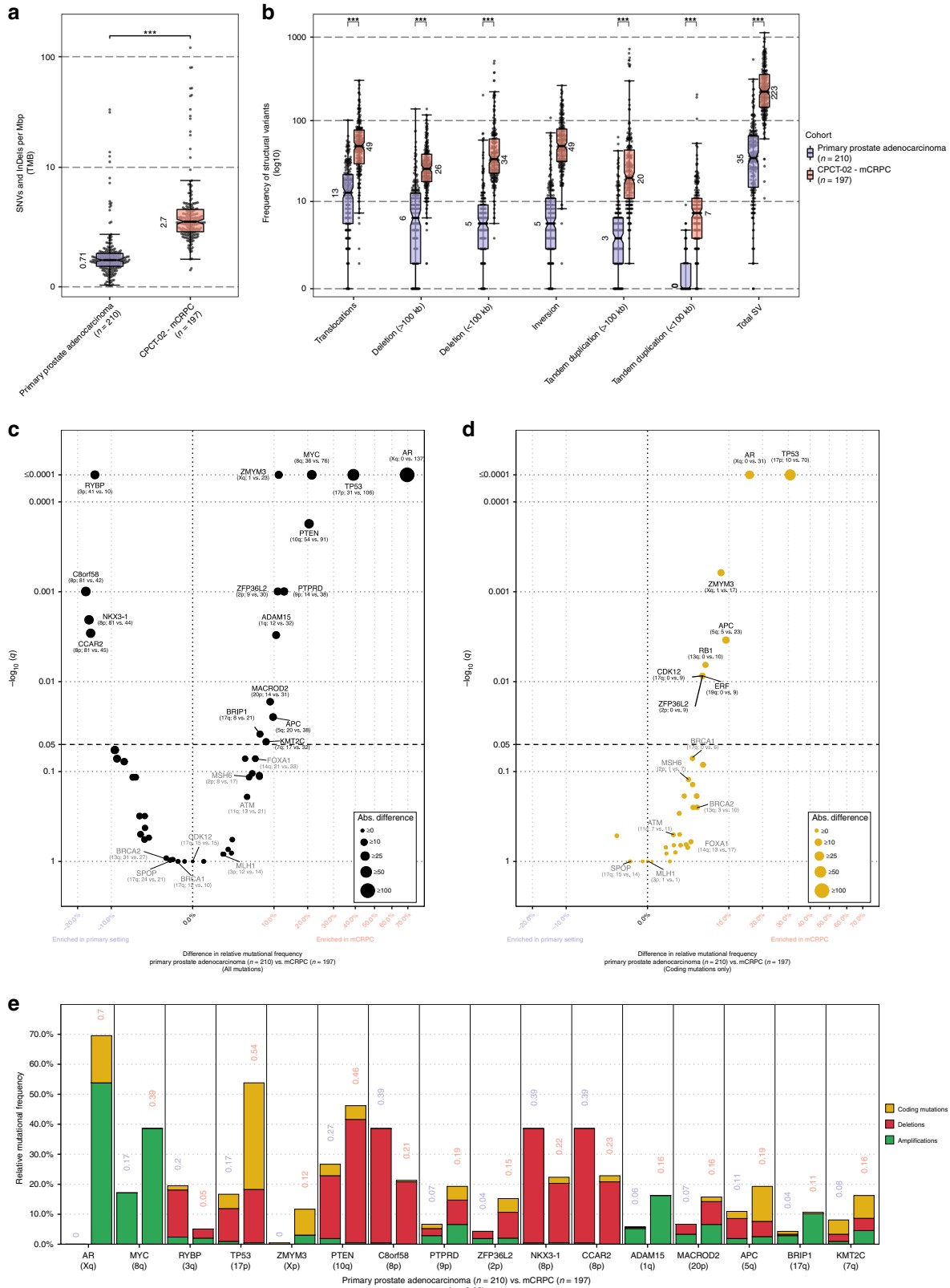

Cluster F was enriched for chromothripsis events, however we could not reproduce a previous finding, suggesting chromothripsis was associated with inversions and p53 inactivation in prostate cancer[21]. Apart from the chromothripsis events, no clear gene aberration was associated with this cluster (Fig. 6b). In the remaining patients, there

were no distinct genomic signatures or biologic rationale for patient clustering (cluster C, E, G, H). In cluster C, conjoint aberrations of *BRCA1* and *TP53* were observed in one patient with a high HR-deficiency prediction score (CHORD), which is known to lead to a small tandem duplication phenotype (<100 kbp)[32]. Two other

**Fig. 3** Comparison of the mutational landscape between primary prostate cancer and mCRPC. **a** Tumor mutational burden (SNVs and InDels per Mbp) from a primary prostate cancer ($n = 210$) and the CPTC-02 mCRPC cohort ($n = 197$). Bee-swarm boxplot with notch of the tumor mutational burden. Boxplot depicts the upper and lower quartiles, with the median shown as a solid line; whiskers indicate 1.5 times the interquartile range (IQR). Data points outside the IQR are shown. Statistical significance was tested with Wilcoxon rank-sum test and $p \leq 0.001$ is indicated as ***. **b** Frequency of structural variant events from an unmatched cohort of primary prostate cancer ($n = 210$) and the CPTC-02 mCRPC cohort ($n = 197$). Boxplot depicts the upper and lower quartiles, with the median shown as a solid line; whiskers indicate 1.5 times the interquartile range (IQR). Data points outside the IQR are shown. Statistical significance was tested with Wilcoxon rank-sum test and $p \leq 0.001$ is indicated as ***. **c** Comparison of the mutational frequencies for driver genes detected by d$N$/d$S$ and/or GISTIC2, or subtype-specific genes, enriched in mCRPC relative to primary prostate cancer or vice-versa. The difference in relative mutational frequency is shown on the $x$-axis and the adjusted $p$-value (two-sided Fisher's Exact Test with BH correction) is shown on the $y$-axis. Size of the dot is proportional to the absolute difference in mutational frequency between both the cohorts. Symbols of genes with $p$-values below 0.05 are depicted in black and additional genes-of-interests are highlighted in gray. The general genomic foci of the gene and absolute number of samples with an aberration per cohort in primary prostate cancer and mCRPC, respectively, is shown below the gene symbol. This analysis was performed on coding mutations, gains and deletions per gene. **d** Same as in **c** but using only coding mutations. **e** Overview of the mutational categories (coding mutations [yellow], deletions [red] and amplifications [green]) of the driver genes detected by d$N$/d$S$ and/or GISTIC2, or subtype-specific genes, enriched in mCRPC relative to primary prostate cancer ($q \leq 0.05$). For each gene the frequency in primary prostate cancer is displayed followed by the frequency in mCRPC

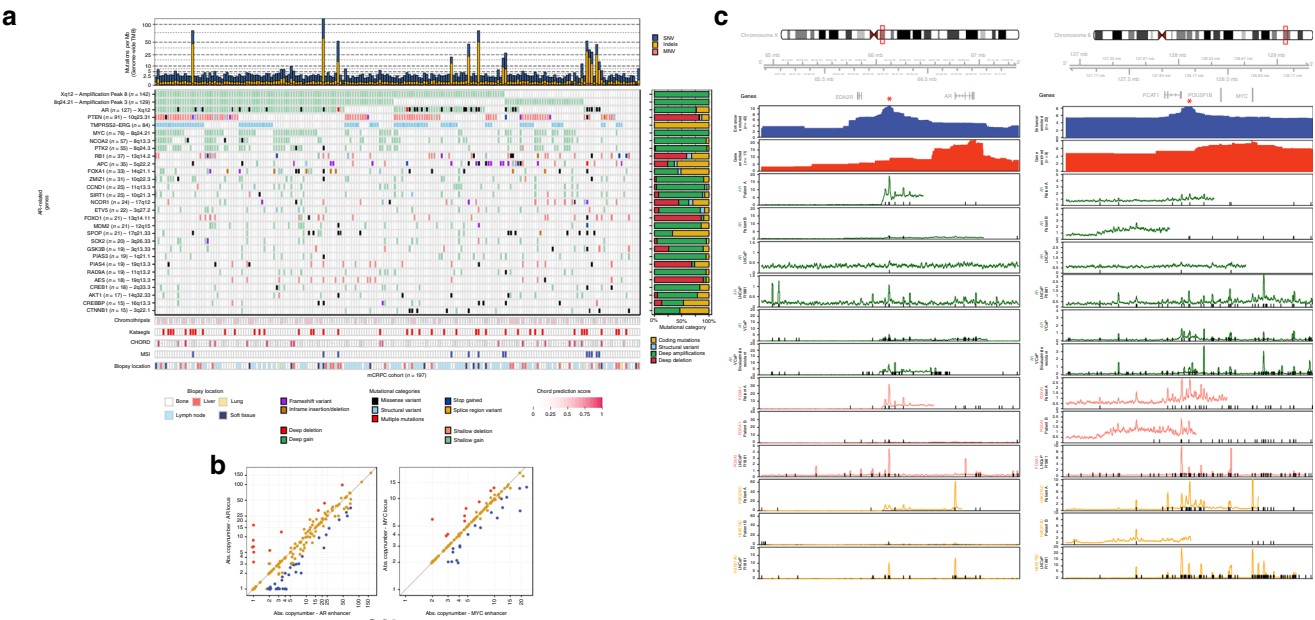

**Fig. 4** WGS reveals novel insight into the various (non-coding) aberrations affecting AR regulation. **a** Mutational overview of top recurrently mutated genes affecting *AR* regulation and their putative enhancer foci (as detected by GISTIC2). The first track represents the number of genomic mutations per Mbp (TMB) per SNV (blue), InDels (yellow), and MNV (orange) category genome-wide (square-root scale). Samples are sorted based on mutual-exclusivity of the depicted genes and foci. The heatmap displays the type of mutation(s) per sample, (light-)green or (light-)red backgrounds depict copy-number aberrations while the inner square depicts the type of (coding) mutation(s). Relative proportions of mutational categories (coding mutations [SNV, InDels and MNV] (yellow), SV (blue), deep amplifications (green), and deep deletions (red)) per gene and foci are shown in the bar plot next to the heatmap. The presence of chromothripsis (light pink), kataegis (red), CHORD prediction score (HR-deficiency) (pink gradient), MSI status (dark blue), and biopsy location are shown as bottom tracks. **b** Overview of the copy-number deviations between putative enhancer and gene regions for *AR* and *MYC*. Samples were categorized as enhancer- (blue) or gene- (red) enriched if enhancer-to-gene ratio deviated >1 studentized residual (residual in standard deviation units) from a 1:1 ratio. **c** Copy number and ChIP-seq profiles surrounding the *AR* and *PCAT1/MYC* gene loci (with 1.25 additional Mbp up-/downstream). The upper panel displays the selected genomic window and the overlapping genes. The first and second track display the aggregated mean copy number (per 0.1 Mbp window) of the enhancer- and gene-enriched samples, respectively. These profiles identify distinct amplified regions (indicated by red asterisk) in proximity to the respective gene bodies. The 3th to 8th tracks represent AR ChIP-seq profiles (median read-coverage per 0.1 Mbp windows) in two mCRPC patients (# 3 and 4), LNCaP (# 5) and LNCaP with R1881 treatment (# 6), VCaP (# 7) and bicalutamide-resistant VCaP (# 8). The 9th to 11th tracks represent FOXA1 ChIP-seq profiles (median read-coverage per 0.1 Mbp windows) in two mCRPC patients (#9 and 10) and LNCaP with R1881 treatment (# 11). The 12th to 14th tracks represent H3K27ac ChIP-seq profiles (median read-coverage per 0.1 Mbp windows) in two mCRPC patients (# 12 and 13) and LNCaP with R1881 treatment (# 14) reflecting active enhancer regions. ChIP-seq peaks (MACS/MACS2; $q < 0.01$) are shown as black lines per respective sample

patients within cluster C displayed a weak CHORD scoring associated with HR-deficiency, however no additional definitive evidence was found for a *BRCA1* loss-of-function mutation within these patients.

In addition to our unsupervised clustering approach, we clustered our samples using the clustering scheme proposed by

TCGA (Supplementary Fig. 13a), which defines seven clusters based on coding mutations and copy-number aberrations in *SPOP*, *FOXA1*, *IDH1*, and *ETS* family gene fusions (and overexpression) per promiscuous partner (*ERG*, *ETV1*, *ETV4*, and *FLI1*)[13]. Unfortunately, we currently lack matched mRNA-sequencing data in our cohort and therefore cannot observe

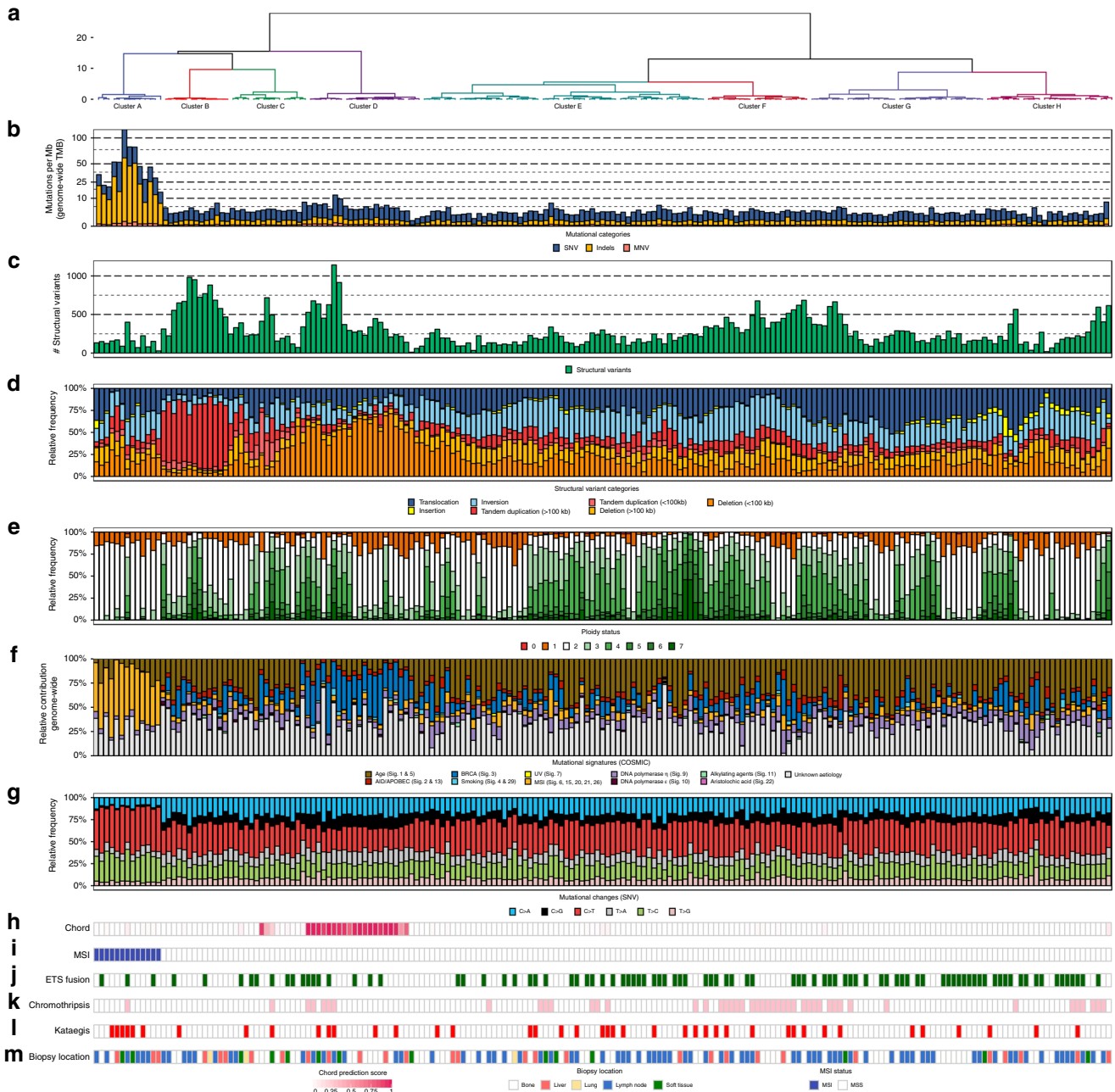

**Fig. 5** Unsupervised clustering of mCRPC reveals distinct genomic phenotypes. **a** Dendrogram of unsupervised clustering with optimal leaf ordering. Top eight clusters are highlighted and denoted based on order of appearance (left to right): A to H. y-axis displays clustering distance (Pearson correlation; ward.D). **b** Number of genomic mutations per Mbp (TMB) per SNV (blue), InDels (yellow), and MNV (orange) category. All genome-wide somatic mutations were taken into consideration (square-root scale). **c** Absolute number of unique structural variants per sample. **d** Relative frequency per structural variant category (translocations, inversions, insertions, tandem duplications, and deletions). Tandem Duplications and Deletions are subdivided into >100 kbp and <100 kbp categories. This track shows if an enrichment for particular category of (somatic) structural variant can be detected, which in turn, can be indicative for a specific mutational aberration. **e** Relative genome-wide ploidy status, ranging from 0 to ≥7 copies. This track shows the relative percentage of the entire genome, which is (partially) deleted (ploidy <2 per diploid genome) or amplified (ploidy >2 per diploid genome). **f** Relative contribution to mutational signatures (COSMIC) summarized per proposed etiology. This track displays the proposed etiology of each SNV based on their mutational contexts. **g** Relative frequency of different SNV mutational changes. **h** HR-deficient prediction score as assessed by CHORD. The binary prediction score of CHORD (ranging from 0 to 1) is shown, in which higher scores reflect more evidence for HR-deficiency in a given sample. **i** MSI status as determined using a stringent threshold of MSI characteristics[40]. **j** Presence of a fusion with a member of the *ETS* transcription factor family. Green color indicates a possible fusion. **k** Presence of chromothripsis. Pink color indicates presence of chromothripsis as estimated by ShatterSeek. **l** Presence of kataegis. Red color indicates presence of one or more regions showing kataegis. **m** General biopsy location

overexpression of fused *ETS* family members, which restricted us to only characterize the genomic breaks of these promiscuous partners. Without incorporation of *ETS* family overexpression, this proposed clustering scheme categorizes 61% of mCRPC into

these seven groups versus 68% of the original cohort containing primary prostate cancer described by TCGA (Supplementary Fig. 13b)[13]. There was no significant correlation between the TCGA clustering scheme and our defined genomic subtypes such

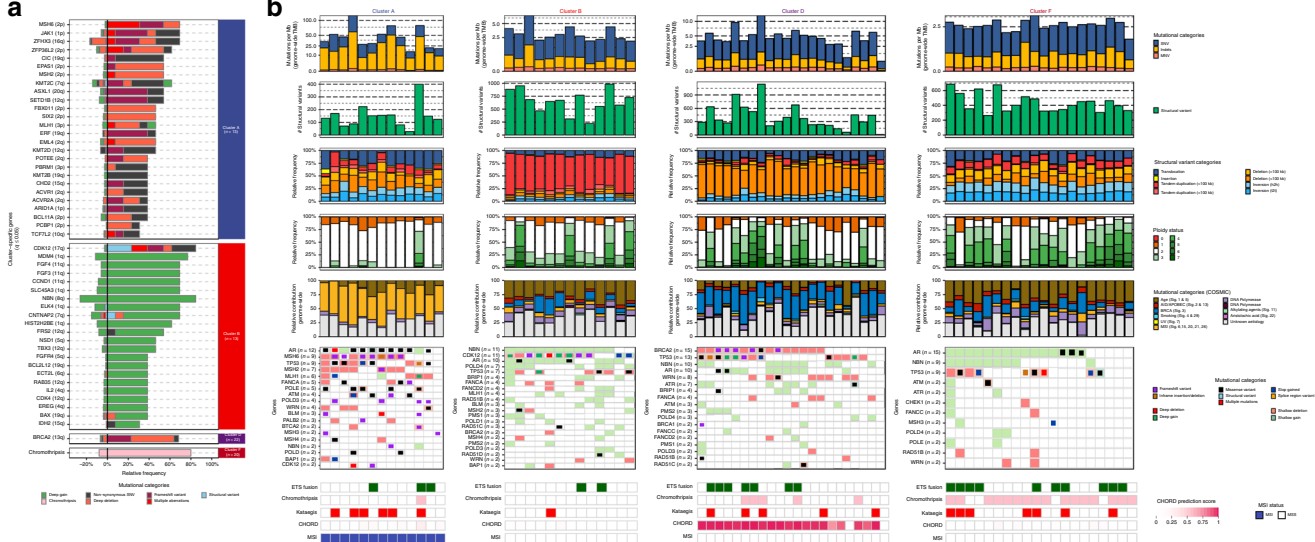

**Fig. 6** Distinct genomic phenotypes in mCRPC are enriched by mutually exclusive aberrations in key pathways. **a** Cluster-specific enrichment of mutated genes (multiple colors), chromothripsis (light pink), and structural variants (light blue) (Fisher's Exact Test with BH correction; $q \leq 0.05$). Percentages to the left of the black line represent the relative mutational frequency in mCRPC samples, which are not present in the respective cluster, while the percentages to the right of the black line represent the relative mutational frequency present in the samples from the tested cluster. **b** Genomic overview with biologically relevant genes in the clusters A, B, D, and F with mutational enrichment of genes or large-scale events. The first track represents the number of genomic mutations per Mbp (TMB) per SNV (blue), InDels (yellow), and MNV (orange) category genome-wide (square-root scale). The second track represents the absolute number of unique structural variants (green) per sample. The third track represents the relative frequency per structural variant category. Tandem duplications and deletions are subdivided into >100 kbp and <100 kbp categories. The fourth track represents relative genome-wide ploidy status, ranging from 0 to ≥7 copies. The fifth track represents the relative contribution to mutational signatures (COSMIC) summarized per proposed etiology. The sixth track displays somatic mutations in the relevant genes found in at least one cluster. The lower tracks represent presence of *ETS* fusions (green), chromothripsis (pink), kataegis (red), CHORD prediction scores (HR-deficiency) (pink gradient), and MSI status (blue) based on a threshold of MSI characteristics

as MSI, BRCAness or $CDK12^{-/-}$. In addition, we did not detect statistical enrichment or depletion ($q \leq 0.05$) between these supervised clusters and additional-mutated genes, kataegis and chromothripsis, only the known enrichment of homozygous *CHD1* deletions in the *SPOP*-cluster was observed[13].

Performing unsupervised clustering and principal component analysis on the primary prostate cancer and metastatic cohorts revealed no striking primary-only genomic subgroup nor did we detect the presence of the mCRPC-derived genomic subgroups in the primary prostate cancer cohort (Supplementary Fig. 14). This could reflect the absence of *CDK12* mutations and the presence of only three sporadic *BRCA2*-mutated samples (1%) in the primary prostate cancer cohort. Furthermore, only one sample (1%) with MSI-like and high TMB (>10), respectively, was observed in the primary cancer cohort. Indeed, there is a striking difference in the mutational load between both disease settings.

## Discussion

We performed WGS of metastatic tumor biopsies and matched-normal blood obtained from 197 patients with mCRPC to provide an overview of the genomic landscape of mCRPC. The size of our cohort enables classification of patients into distinct disease subgroups using unsupervised clustering. Our data suggest that classification of patients using genomic events, as detected by WGS, improves patient stratification, specifically for clinically actionable subgroups such as BRCA-deficient and MSI patients. Furthermore, we confirm the central role of AR signaling in mCRPC that mediates its effect through regulators located in non-coding regions and the apparent difference in primary versus metastatic prostate cancers.

The classification of patients using WGS has the advantage of being, in theory, more precise in determining genomically defined subgroups in prostate cancer compared to analyses using targeted panels consisting of a limited number of genes, or exome sequencing. The identification of subgroups based on predominant phenotypic characteristics encompassing genomic signatures may be clinically relevant and our clustering analysis refines patient classification. In cluster A, we observed a high TMB, which has been associated in other tumor types with a high sensitivity to immune check-point inhibitors[9,11,12]. Clinical trials using pembrolizumab in selected mCRPC patients are underway (KEYNOTE-028, KEYNOTE-199)[36,37]. Interestingly, in both cluster B and cluster D, we identified patients that did not have the defining biallelic *CDK12* or *BRCA2* (somatic) mutation. Such patients might be deemed false-negatives when using FDA-approved assays (BRCAnalysis™ and FoundationFocus™), currently used in breast cancer diagnosis and based on the presence of *BRCA1/2* mutations, to predict response to poly(ADP-ribose) polymerase (PARP) inhibitors and/or platinum compounds. The first clinical trials combining PARP inhibitors with AR-targeted therapies in mCRPC show promising results[8]. Thus, WGS-based stratification may improve the patient classification of DNA repair-deficient tumors as it uses the genome-wide scars caused by defective DNA repair to identify tumors that have these deficiencies.

The use of WGS also allowed us to gain more insight into the role of non-coding regions of the genome in prostate cancer. We confirmed the amplification of a recently reported *AR*-enhancer[20,21,30]. In line with the cell line-based observations, we show AR binding at these mCRPC-specific enhancer regions, providing the first clinical indication that *AR*-enhancer

**Table 1 Overview of the distinctive characteristics for each cluster (A-H)**

| | Number of patients (n; % of cohort) | Tumor mutational burden (CDS) | SNV/InDel ratio | Number of structural variants | Main structural variant category or differentiating category | Ploidy status | Main mutational signature | Top 3 cluster-specific aberrations (% of cluster) | ETS-fusions (n) | Chromothripsis (n) | Kataegis (n) |
|---|---|---|---|---|---|---|---|---|---|---|---|
| Cluster A | 13 (6.6) | 36,88 | 0,99 | 149 | None | 1,92 | MSI | MSH6 (69.2), JAK1 (69.2), CIC (58.3) | 3 | 1 | 6 |
| Cluster B | 13 (6.6) | 2,44 | 7,07 | 669 | Tandem duplications (>100 kb) | 2,39 | N/A | CDK12 (84.6), FGF3 (69.2), FGF4 (69.2) | 2 | 0 | 1 |
| Cluster C | 15 (7.6) | 3,00 | 6,73 | 237 | Tandem duplications (<100 kb) | 3,19 | N/A | None | 7 | 1 | 2 |
| Cluster D | 22 (11.2) | 4,39 | 7,28 | 323 | Deletions (>100 kb) | 2,16 | BRCA | BRCA2 (68.2) | 7 | 5 | 5 |
| Cluster E | 55 (27.9) | 2,12 | 7,13 | 178 | None | 3,24 | N/A | None | 25 | 8 | 13 |
| Cluster F | 20 (10.2) | 2,51 | 6,15 | 400 | None | 3,35 | N/A | Chromothripsis (80,0) | 10 | 16 | 7 |
| Cluster G | 34 (17.3) | 2,12 | 6,13 | 222 | None | 2,98 | N/A | None | 23 | 8 | 5 |
| Cluster H | 25 (12.7) | 2,30 | 5,81 | 201 | Insertions | 1,97 | N/A | None | 16 | 7 | 3 |

All numbers are median of the cluster, unless otherwise indicated
CDS coding sequence

amplification also increases AR signaling in mCRPC tumors. These findings are supported by previous studies demonstrating that this amplification ultimately resulted in significantly elevated expression of *AR* itself[20,21,30]. Furthermore, we confirm a recurrent focal amplification near *PCAT1*, which shows robust chromatin binding for AR in mCRPC samples, providing clinical proof-of-concept of a functional enhancer that is also active and AR-bound in cell line models. Recent research elucidated to the functional importance of this region in regulating *MYC* expression in prostate cancer, which could highlight a putative role of this somatically acquired amplification[31]. However, the WGS and ChIP-seq data presented here are not conclusive in elucidating the definitive role of this amplified region in regulating *MYC* expression and further mechanistic studies are needed to establish a potential link to *MYC* regulation.

In addition, *PCAT1* is a long non-coding RNA, which is known to be upregulated in prostate cancer and negatively regulates *BRCA2* expression while positively affecting *MYC* expression[38,39]. Combining our WGS approach with AR, FOXA1, and H3K27ac ChIP-seq data, we identify non-coding regions affecting both *AR* itself, and possibly *MYC*, through *AR*-enhancer amplification as a potential mechanism contributing to castration resistance.

A potential pitfall of our clustering analysis is the selection of features used; for this we made a number of assumptions based on the literature and distribution of the structural variants within our cohort[19–21,32]. As the input of features and weights for clustering analysis are inherent to the clustering outcome, we performed additional clustering analyses using various combinations of these features and applied alternative approaches but did not detect striking differences compared to the current approach. Another potential pitfall of the employed hierarchical clustering scheme is that patients are only attributed to a single cluster. An example of this can be seen in cluster A where a patient is grouped based on its predominant genotype (MSI) and associated mutations in MMR-related genes (*MLH1*, *POLE*, *POLD3*, and *BLM*), but this sample also displays an increased number of structural variants and increased ploidy status and harbors a pathogenic *BRCA2* mutation. However, it is missing the characteristic number of genomic deletions (<100 kbp) and BRCA mutational signature associated with $BRCA2^{-/-}$ samples that define cluster D. Despite these pitfalls we conclude that unbiased clustering contributes towards improved classification of patients.

The CPCT-02 study was designed to examine the correlation of genomic data with treatment outcome after biopsy at varying stages of disease. Our cohort contains patients with highly variable pre-treatment history and since the treatments for mCRPC patients nowadays significantly impacts overall survival, the prognosis of patients differs greatly. Therefore, correlation between genomic data and clinical endpoints, such as survival is inherently flawed due to the very heterogeneous nature of the patient population. Moreover, our analysis comparing primary and metastatic samples shows a significant increase in the number of genomic aberrations with advancing disease, meaning that the difference in timing of the biopsies may bias the prognostic value of the data. In future studies, we plan to gather all known clinically defined prognostic information and determine whether the genomic subtypes increase the ability to predict outcome. Unfortunately, some clinical parameters with prognostic importance such as ethnicity will not be available due to ethical regulations. Moreover, we will increase the sample size, in order to correlate genomic features to clinical parameters to better determine whether the subtypes we identified are stable over time. Therefore, we are currently unable to present meaningful correlations between clinical endpoints and the clusters we identified.

Overall, we show the added value of WGS-based unsupervised clustering in identifying patients with genomic scars who are eligible for specific therapies. Since our clustering method does not rely on one specific genetic mutation we are able to classify patients even when WGS (or our methodology) does not find conclusive evidence for (biallelic) mutations in the proposed gene-of-interest. Further research should validate clinical response and outcome on specific therapies in matched sub-groups. This study also shows that a large population of mCRPC patients do not fall into an as-of-yet clinically relevant or biologically clear genotype and further research can help elucidate the oncogenic driver events and provide new therapeutic options.

## Methods

**Patient cohort and study procedures.** Patients with metastatic prostate cancer were recruited under the study protocol (NCT01855477) of the Center for Personalized Cancer Treatment (CPCT). This consortium consists of 41 hospitals in The Netherlands (Supplementary Table 1). This CPCT-02 protocol was approved by the medical ethical committee (METC) of the University Medical Center Utrecht and was conducted in accordance with the Declaration of Helsinki. Patients were eligible for inclusion if the following criteria were met: (1) age ≥ 18 years; (2) locally advanced or metastatic solid tumor; (3) indication for new line of systemic treatment with registered anti-cancer agents; (4) safe biopsy according to the intervening physician. For the current study, patients were included for biopsy between 03 May 2016 and 28 May 2018. Data were excluded of patients with the following characteristics: (1) hormone-sensitive prostate cancer; (2) neuro-endocrine prostate cancer (as assessed by routine diagnostics); (3) unknown disease status; (4) prostate biopsy (Fig. 1a). All patients provided written informed consent before any study procedure. The study procedures consisted of the collection of matched peripheral blood samples for reference DNA and image-guided percutaneous biopsy of a single metastatic lesion. Soft tissue lesions were biopsied preferentially over bone lesions. The clinical data provided by CPCT have been locked at 1st of July 2018.

**Collection and sequencing of samples.** Blood samples were collected in CellSave preservative tubes (Menarini-Silicon Biosystems, Huntington Valley, PA, USA) and shipped by room temperature to the central sequencing facility at the Hartwig Medical Foundation[40]. Tumor samples were fresh-frozen in liquid nitrogen directly after the procedure and send to a central pathology tissue facility. Tumor cellularity was estimated by assessing a hematoxylin-eosin (HE) stained 6 micron thick section. Subsequently, 25 sections of 20 micron were collected for DNA isolation. DNA was isolated with an automated workflow (QiaSymphony) using the DSP DNA Midi kit for blood and QiaSymphony DSP DNA Mini kit for tumor samples according to the manufacturer's protocol (Qiagen). DNA concentration was measured by Qubit™ fluorometric quantitation (Invitrogen, Life Technologies, Carlsbad, CA, USA). DNA libraries for Illumina sequencing were generated from 50 to 100 ng of genomic DNA using standard protocols (Illumina, San Diego, CA, USA) and subsequently whole-genome sequenced in a HiSeq X Ten system using the paired-end sequencing protocol (2 × 150 bp). Whole-genome alignment (GRCh37), somatic variants (SNV, InDel (max. 50 bp), MNV), structural variant and copy number calling and in silico tumor cell percentage estimation were performed in a uniform manner as detailed by Priestley et al.[40]. Mean read coverages of reference and tumor BAM were calculated using Picard Tools (v1.141; CollectWgsMetrics) based on GRCh37[41].

**Additional annotation of somatic variants and heuristic filtering.** In addition, heuristic filtering removed somatic SNV, InDel, and MNV variants based on the following criteria: (1) minimal alternative reads observations ≤ 3; (2) gnomAD exome (ALL) allele frequency ≥ 0.001 (corresponding to ~62 gnomAD individuals); and (3) gnomAD genome (ALL) ≥0.005 (~75 gnomAD individuals)[42]. gnomAD database v2.0.2 was used. Per gene overlapping a genomic variant, the most deleterious mutation was used to annotate the overlapping gene. Structural variants, with BAF ≥0.1, were further annotated by retrieving overlapping and nearest up- and downstream annotations using custom R scripts based on GRCh37 canonical UCSC promoter and gene annotations with respect to their respective up- or downstream orientation (if known)[43]. Only potential fusions with only two different gene-partners were considered (e.g., *TMPRSS2-ERG*); structural variants with both breakpoints falling within the same gene were simply annotated as structural variant mutations. Fusion annotation from the COSMIC (v85), CGI and CIVIC databases were used to assess known fusions[44–46]. The COSMIC (v85), OncoKB (July 12, 2018), CIVIC (July 26, 2018), CGI (July 26, 2018) and the list from Martincorena et al.[26] (dN/dS) were used to classify known oncogenic or cancer-associated genes[44–46].

**Ploidy and copy-number analysis.** Ploidy and copy-number (CN) analysis was performed by a custom pipeline as detailed by Priestley et al.[40]. Briefly, this pipeline combines B-allele frequency (BAF), read depth, and structural variants to estimate the purity and CN profile of a tumor sample. Recurrent focal and broad CN alterations were identified by GISTIC2.0 (v2.0.23)[27]. GISTIC2.0 was run with the following parameters: (a) genegistic 1; (b) gcm extreme; (c) maxseg 4000; (d) broad 1; (e) brlen 0.98; (f) conf 0.95; (g) rx 0; (h) cap 3; (i) saveseg 0; (j) armpeel 1; (k) smallmem 0; (l) res 0.01; (m) ta 0.1; (n) td 0.1; (o) savedata 0; (p) savegene 1; (q) gvt 0.1. Categorization of shallow and deep CN aberration per gene was based on thresholded GISTIC2 calls. Focal peaks detected by GISTIC2 were re-annotated, based on overlapping genomic coordinates, using custom R scripts and UCSC gene annotations. GISTIC2 peaks were annotated with all overlapping canonical UCSC genes within the wide peak limits. If a GISTIC2 peak overlapped with ≤3 genes, the most-likely targeted gene was selected based on oncogenic or tumor-suppressor annotation in the COSMIC (v85), OncoKB (July 12, 2018), CIVIC (July 26, 2018), and CGI (July 26, 2018) lists[26,44–46]. Peaks in gene deserts were annotated with their nearest gene.

**Estimation of tumor mutational burden.** The mutation rate per megabase (Mbp) of genomic DNA was calculated as the total genome-wide amount of SNV, MNV, and InDels divided over the total amount of callable nucleotides (ACTG) in the human reference genome (hg19) FASTA sequence file:

$$\text{TMB}_{genomic} = \frac{\left(\text{SNV}_g + \text{MNV}_g + \text{InDels}_g\right)}{\left(\frac{2858674662}{10^6}\right)} \quad (1)$$

The mutation rate per Mbp of coding mutations was calculated as the amount of coding SNV, MNV, and InDels divided over the summed lengths of distinct non-overlapping coding regions, as determined on the subset of protein-coding and fully supported (TSL = 21) transcripts in GenCode v28 (hg19)[47]:

$$\text{TMB}_{coding} = \frac{\left(\text{SNV}_c + \text{MNV}_c + \text{InDels}_c\right)}{\left(\frac{28711682}{10^6}\right)} \quad (2)$$

**MSI and HR-deficiency prediction.** HR-deficiency/BRCAness was estimated using the CHORD classifier (Nguyen, van Hoeck and Cuppen, manuscript in preparation). This classifier was based on the HRDetect[48] algorithm, however, redesigned to improve its performance beyond primary BC. The binary prediction score (ranging from 0 to 1) was used to indicate BRCAness level within a sample. To elucidate the potential target gene(s) in the HR-deficient samples (Fig. 4), we used the list of BRCAness genes from Lord et al.[35].

MSI status was determined based on the following criteria: if a sample contained >11,436 genomic InDels (max. 50 bp, with repeat-stretches of ≥4 bases, repeat length sequence between 2 and 4, or if these InDels consist of a single repeat sequence, which repeats ≥5 times), the sample was designated as MSI[40].

**Detection of (onco-)genes under selective pressure.** To detect (onco-)genes under tumor-evolutionary mutational selection, we employed a Poisson-based dN/dS model (192 rate parameters; under the full trinucleotide model) by the R package dndscv (v0.0.0.9)[26]. Briefly, this model tests the normalized ratio of non-synonymous (missense, nonsense, and splicing) over background (synonymous) mutations while correcting for sequence composition and mutational signatures. A global q-value ≤ 0.1 (with and without taking InDels into consideration) was used to identify statistically significant (novel) driver genes.

**Identification of hypermutated foci (kataegis).** Putative kataegis events were detected using a dynamic programming algorithm, which determines a globally optimal fit of a piecewise constant expression profile along genomic coordinates as described by Huber et al.[49] and implemented in the tilingarray R package (v1.56.0). Only SNVs were used in detecting kataegis. Each chromosome was assessed separately and the maximum number of segmental breakpoints was based on a maximum of five consecutive SNVs (max. 5000 segments per chromosome). Fitting was performed on $\log_{10}$-transformed intermutational distances. Per segment, it was assessed if the mean intermutational distance was ≤2000 bp and at least five SNVs were used in the generation of the segment. A single sample with >200 distinct observed events was set to zero observed events as this sample was found to be hypermutated throughout the entire genome rather than locally. Kataegis was visualized using the R package karyoploteR (v1.4.1)[50].

**Mutational signatures analysis.** Mutational signatures analysis was performed using the MutationalPatterns R package (v1.4.2)[51]. The 30 consensus mutational signatures, as established by Alexandrov et. al, (matrix S*ij*; $i = 96$; number of tri-nucleotide motifs; $j = 30$; number of signatures) were downloaded from COSMIC (as visited on 23-05-2018)[29]. Mutations (SNVs) were categorized according to their respective trinucleotide context (hg19) into a mutational spectrum matrix M*ij* ($i = 96$; number of trinucleotide contexts; $j = 196$; number of samples) and subsequently, per sample a constrained linear combination of the thirty consensus mutational signatures was constructed using non-negative least squares regression implemented in the R package pracma (v1.9.3).

Between two and 15 custom signatures were assessed using the NMF package (v0.21.0) with 1000 iterations[52]. By comparing the cophenetic correlation coefficient, residual sum of squares and silhouette, we opted to generate five custom

signatures. Custom signatures were correlated to existing (COSMIC) signatures using cosine similarity.

**Detection of chromothripsis-like events.** Rounded absolute copy number (excluded Y chromosome) and structural variants (BAF ≥ 0.1) were used in the detection of chromothripsis-like events by the Shatterseek software (v0.4) using default parameters[53]. As a precise standardized definition of chromothripsis has not yet been fully established, and as per the author's instruction, we performed visual inspection of reported chromothripsis-like events after dynamically adapting criteria thresholds (taking the recommended thresholds into consideration). We opted to use the following criteria: (a) Total number of intrachromosomal structural variants involved in the event ≥25; (b) max. number of oscillating CN segments (two states) ≥7 or max. number of oscillating CN segments (three states) ≥14; (c) Total size of chromothripsis event ≥20 Mbp; (d) Satisfying the test of equal distribution of SV types ($p > 0.05$); and (e) Satisfying the test of non-random SV distribution within the cluster region or chromosome ($p \leq 0.05$).

**Unsupervised clustering of mCRPC WGS characteristics.** Samples were clustered using the Euclidian distance of the Pearson correlation coefficient ($1 - r$) and Ward. D hierarchical clustering based on five basic whole-genome characteristics; number of mutations per genomic Mbp (SNV, InDel, and MNV), mean genome-wide ploidy, number of structural variants and the relative frequencies of structural variant categories (inversions, tandem duplications (larger and smaller than 100 kbp), deletions (larger and smaller than 100 kbp), insertions and inter-chromosomal translocations). Data was scaled but not centered (root mean square) prior to calculating Pearson correlation coefficients. After clustering, optimal leaf ordering (OLO) was performed using the seriation package (v1.2.3)[54]. The elbow method was employed to determine optimal number of discriminating clusters (Supplementary Fig. 10) using the factoextra package (v1.0.5). Bootstrapping was performed using the pvclust package (v2.0) with 5000 iterations.

Cluster-specific enrichment of aberrant genes (either through SV, deep copy-number alteration, or coding SNV/InDel/MNV), kataegis, chromothripsis, GISTIC2 peaks, and predicted fusions between clusters was tested using a two-sided Fisher's Exact Test and Benjamini–Hochberg correction.

A principal component analysis (with scaling and centering) using the prcomp R package[55] was performed on the chosen genomic features and $\cos^2$ values for each feature per principal component were retrieved to determine the importance of each feature per respective principal component.

To test the robustness of our clustering, we performed unsupervised clustering, and also other techniques, using various combinations of structural variants and clustering mechanisms as a surrogate for different genome-instability metrics but this analysis did not reveal any striking new clusters.

**Supervised clustering based on mutually exclusive aberrations.** Samples were sorted on mutual-exclusivity of *SPOP*, *FOXA1*, and *IDH1* coding mutations and copy-number aberrations and *ETS* family gene fusions (and overexpression) per promiscuous partner (*ERG*, *ETV1*, *ETV4*, and *FLI1*) as defined in primary prostate cancer[13]. Supplementary Table S1A of the article "The Molecular Taxonomy of Primary Prostate Cancer"[13] was used to determine the relative frequency and mutational types of each of the respective primary prostate cancer within the TCGA cohort. In addition, as the TCGA cohort did not denote high-level/deep amplifications, we did not incorporate these either in this analysis.

**Correlation of the detection rate of genomic aberrations versus tumor cell percentages.** Absolute counts of SNV, InDels, MNV and SV were correlated to the in silico estimated tumor cell percentage using Spearman's correlation coefficient.

**Correlation of pre-treatment history with detected aberrations and WGS characteristics.** Pre-treatment history of patients was summarized into ten groups:

1. Only chemo-treatment (with radio-nucleotides).
2. Only chemo-treatment (without radio-nucleotides).
3. Only radio-nucleotides.
4. Only secondary anti-hormonal therapy (with radio-nucleotides).
5. Only secondary anti-hormonal therapy (without radio-nucleotides).
6. Secondary anti-hormonal therapy + one chemo-treatment (with radio-nucleotides)
7. Secondary anti-hormonal therapy + two chemo-treatment (with radio-nucleotides)
8. Secondary anti-hormonal therapy + one chemo-treatment (without radio-nucleotides)
9. Secondary anti-hormonal therapy + two chemo-treatment (without radio-nucleotides)
10. No additional treatment after androgen deprivation therapy.

Association with mutated genes, presence of chromothripsis, presence of kataegis, MSI-status, and genomic subtypes was tested with a two-sided Fisher's exact test with Benjamini–Hochberg correction.

**ChIP-seq experimental set-up and analysis.** *ChIP-seq cell culturing*: VCaP cells were incubated in RPMI medium in additional with 10% fetal bovine serum (FBS). Bicalutamide-resistant VCaP cells (VCaP-Bic) were cultured in RPMI medium supplemented with 10% dextran charcoal-stripped bovine serum (DCC) and 10-6M bicalutamide. VCaP cells were hormone deprived in RPMI medium with 10% DCC for 3 days before the ChIP-seq experiment.

*ChIP-seq and peak calling analysis*: For both cell and tissue ChIPs, 5 µg of antibody and 50 µg of magnetic protein A or G beads (10008D or 10009D, Thermo Fisher Scientific) were used per IP. The following antibodies were used: Foxa1/2 (M-20, sc-6554 Santa Cruz Biotechnology), AR (N-20, sc-816 Santa Cruz Biotechnology), and H3K27ac (39133, Active Motif). ChIP-seq was performed as described previously[56]. In brief, fresh-frozen tissue was cryosectioned into 30 micron thick slices and stored at −80 °C till processing. Samples were fixed using 2 mM DSG (20593; Thermo Fisher Scientific) in solution A (50 mM Hepes-KOH, 100 mM NaCl, 1 mM EDTA, 0.5 mM EGTA) while rotating for 25 min at room temperature, followed by the addition of 1% formaldehyde and another 20 min incubation at room temperature. The reaction was quenched by adding a surplus of glycine. Subsequently, tissue sections were pelleted and washed with cold PBS. Tissue was disrupted using a motorized pellet pestle (Sigma-Aldrich) to disrupt the tissue in cold PBS and obtain a cell suspension, after which the nuclei were isolated and the chromatin was sheared. During immunoprecipitation, human control RNA (4307281; Thermo Fisher Scientific) and recombinant Histone 2B (M2505S; New England Biolabs) were added as carriers, as described previously[57].

Immunoprecipitated DNA was processed for sequencing using standard protocols and sequenced on an Illumina HiSeq 2500 with 65 bp single end reads. Sequenced samples were aligned to the reference human genome (Ensembl release 55: Homo sapiens GRCh 37.55) using Burrows-Wheeler Aligner (BWA, v0.5.10)[58], reads with a mapping quality >20 were used for further downstream analysis.

For the tissues, peak calling was performed using MACS2[59] with option --nomodel. In addition, peaks were called against matched input using DFilter[60] in the refine setting with a bandwidth of 50 and a kernel size of 30. Only peaks that were shared between the two algorithms were considered.

For the cell lines, peaks were obtained with MACS (v1.4; $p \leq 10^{-7}$).

The AR and FOXA1 ChIP-seq data for LNCAP with/-out R1881 was obtained from GSE94682[61]. The H3K27ac ChIP-seq data for LNCAP was obtained from GSE114737[56].

*Determining enrichment of enhancer to gene ratios*: Absolute copy-numbers segments overlapping the gene loci and putative enhancer region (as detected by GISTIC2; focal amplification peaks with a width <5000 bp) were retrieved per sample. If regions overlapped multiple distinct copy-number segments, the maximum copy-number value of the overlapping segments was used to represent the region. Samples with gene-to-enhancer ratios deviating >1 studentized residual from equal 1:1 gene-to-enhancer ratios (linear model: $\log_2$(copy number of enhancer) – $\log_2$(copy number of gene locus) ~ 0) were categorized as gene or enhancer enriched. Based on the direction of the ratio, samples were either denoted as enhancer (if positive ratio) or gene (if negative ratio) enriched.

**Comparison of unmatched primary prostate cancer and mCRPC.** Mutational frequencies of the drivers (dN/dS and or GISTIC2) and subtype-specific genes were compared to a separate (unmatched) cohort of primary prostate cancer ($n = 210$) focusing on Gleason score (GS) of $3 + 3$, $3 + 4$, or $4 + 3$, as described by Fraser et al.[15] and Espiritu et al.[25]. Briefly, whole-genome sequencing reads were mapped to the human reference genome (GRCh37) using BWA[58] (v0.5.7) and downstream analysis was performed using Strelka[62] (v.1.0.12) for mutational calling using a matched-normal design (SNVs and InDels), copy-number alterations were estimated with TITAN[63] (v1.11.0), and SNP array data as described in Espiritu et al.[25] with Delly[64] (v0.5.5 and v0.7.8) was used for detecting structural variants (translocations, inversions, tandem duplications, and deletions). Large insertion calls and overall ploidy was not available for the primary prostate cancer cohort.

Tumor mutational burden (TMB) was calculated by dividing the number of SNVs and InDels by the total amount of callable bases in the human reference genome (GRCh37), identical to Eq. 1. MNV calls were not available for the primary prostate cancer cohort.

Multiple aberrations per gene within a sample were summarized as a single mutational event, e.g., a deletion and mutation in *PTEN* would only count for a single mutation in the sample. Only non-synonymous mutations and gains/deletions overlapping with coding regions were used. Statistically significant differences in mutational frequencies were calculated using a two-sided Fisher's Exact test with Benjamini–Hochberg correction.

The primary prostate cancer dataset was clustered together with the mCRPC cohort using the Euclidian distance of the Pearson correlation coefficient ($1 - r$) and Ward.D hierarchical clustering based on three basic whole-genome characteristics, which were available for all samples; number of mutations per genomic Mbp (SNVs and InDels), number of structural variants and the relative frequencies of structural variant categories (inversions, tandem duplications (larger and smaller than 100 kbp), deletions (larger and smaller than 100 kbp), and interchromosomal translocations).

**Reporting summary**. Further information on research design is available in the Nature Research Reporting Summary linked to this article.

## Data availability

The data that support the findings of this study are available from Hartwig Medical Foundation, which were used under data request number DR-011 for the current study. Both WGS and clinical data are freely available for academic use from the Hartwig Medical Foundation through standardized procedures and request forms can be found at https://www.hartwigmedicalfoundation.nl[40]. The ChIP-seq profiles (aligned reads and MACS/MACS2 peaks) as analyzed and shown in this manuscript have been deposited on GEO under accession number: GSE138168.

## Code availability

All tools and scripts used for processing of the WGS data are available at https://github.com/hartwigmedical/ and/or can be provided by authors upon request.

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

## Acknowledgements

This publication and the underlying study have been made possible in parts by the data that the Hartwig Medical Foundation and the Center of Personalized Cancer Treatment (CPCT) have made available to the study. We would like to thank the local principal investigators of all contributing centers for their help with patient enrollment (listed in Supplementary Table 1). We would also like to thank Tesa M. Severson for her help with the computational analyses of the ChIP-seq data, Suzan Stelloo for providing ChIP-seq results on cell lines and Arne van Hoeck for providing the CHORD (HR-deficiency) prediction scores. We thank Dr. Joost van Rosmalen for his advises on the statistical analyses. In addition, we would like to thank the Barcode for Life foundation for making this research possible. Figure 1 was created with BioRender.com. This work was supported in parts by a KWF-Alpe d'HuZes project [NKI 2014-7080], a grant from Astellas Pharma [Lolkema/NL-72-RG-11] and a Johnson & Johnson grant [212082PCR3014]. This work was supported by the NIH/NCI under award number P30CA016042. H.J.G.v.D.W., J.v.R. and the Erasmus MC Cancer Computational Biology Center (CCBC) were financed through a grant from the Daniel den Hoed foundation.

## Author contributions

L.F.v.D., J.v.R., M.S. M.P.L. and H.J.G.v.d.W. wrote the manuscript, which all authors critically reviewed. J.v.R and H.J.G.v.d.W. performed the bioinformatics analyses and visualization. L.F.v.D., M.P.L. and N.M. managed clinical data assessment. Y.Z. and W.Z. performed ChIP-seq experiments and data analyses. N.M., M.P.L. P.H., A.M.B., M.S.v.d.H., I.M.v.O. and R.d.W. are clinical contributors. M.P.L., M.S.v.d.H., E.V., N.S., J.W.M.M. and S.S. are part of the CPCT-02 study. E.P.J.G.C. coordinated the sequencing of samples and contributed to the bioinformatics analyses. T.N.Y., J.L. and P.C.B. assisted with acquisition of the primary prostate cancer cohort and subsequent analysis.

## Competing interests

The authors declare no competing interests.
