## [Peer Review File · Nature Communications]

Reviewers' comments:

Reviewer #1 (Remarks to the Author):

This manuscript by van Dessel, van Riet et al. describes the use of whole-genome sequencing to categorize 197 cases of metastatic, castration-resistant prostate cancer (CRPC) into 8 clusters based on ploidy, mutational burden, and the number and relative frequencies of insertions, deletions, tandem duplications and translocations. 4 of those clusters correspond approximately to known patterns of genomic instability in prostate cancer linked to an underlying cause (biallelic BRCA2 or BRCA1 deficiency, biallelic CDK12 deficiency, microsatellite instability, or chromothripsis). The other 4 clusters exhibit genomic signatures that proved more difficult to associate with a known subtype or single pathogenic event.

The study's concordance with the body of literature on prostate cancer genomic instability and molecular subtypes is a good sign that this whole-genome sequencing methodology will be clinically useful for stratifying some patients with metastatic CRPC into categories associated with different prognoses or outcomes to specific treatments. On the other hand, the study's ability to provide additional insights to the field regarding outcomes associated with these 8 clusters is limited by the few clinical details linked to these samples (biopsy site, age, and previous treatments undergone by the patients) and their retrospective rather than prospective nature. There are opportunities for the authors to perform additional informatics analyses to better integrate their findings with similar studies and draw conclusions about which known subtypes preferentially progress to the metastatic CRPC stage. For example, how well-represented are each of these 8 clusters in the localized or androgen-dependent stages of prostate cancer? And how do they overlap with known molecular subtypes?

The study includes other experiments that build on the whole genome sequencing data, particularly ChIP-seq studies that yield some interesting findings such as the validation that certain amplification events observed in patients coincide with genome binding sites for the AR and FOXA1 transcription factors. However, these data are different enough from the title of the manuscript and the theme of demonstrating the clinical utility of whole genome sequencing data that they do not help the study form a single cohesive story. Overall, the study provides a valuable analysis of a large number of metastatic, castration-resistant prostate cancer samples, with subtyping based on somatically acquired genomic changes. Specific comments and suggestions are below.

Comments

Major Comment #1: The clinical details linked to the new samples seem to be biopsy site, age, and previous treatments of the patients, and do not include prospectively collected data. This limits the study's ability to associate their 8 clusters/subtypes with distinct outcomes, as the most important clinical outcome here is the already-achieved status of these samples as metastatic CRPC. These data might be best used to demonstrate the enrichment of certain subtypes among metastatic CRPC and the enhanced ability of certain subtypes of early prostate cancer to progress to this stage. As a result, it may be important to perform a more thorough analysis of other prostate cancer whole genome sequencing datasets for comparison to this metastatic CRPC group. An example of how the authors have already started to do this is their finding that their metastatic CRPC samples have an average tumor mutation burden in coding regions more than 2-fold higher than primary prostate cancer (lines 64-68).

There are several opportunities for additional informatics analyses to better integrate the new findings with similar studies. For example, one of the strengths of the manuscript (the relatively large number of metastatic CRPC biopsies) might be better exhibited by making more comparisons to publicly available WGS data from localized or androgen-dependent disease. Using the clustering criteria identified in this study (ploidy, mutational burden, and the number and relative frequencies

of insertions, deletions, tandem duplications and translocations), can the authors analyze public whole-genome sequencing data to determine how well-represented are each of these 8 clusters in the localized CRPC or androgen-dependent stages of prostate cancer? Supplementary figure 2 contains some comparison of tumor mutation burden in metastatic CRPC samples vs. localized CRPC samples. It would be very valuable if the authors expanded this line of investigation to categorize the localized CRPC samples into clusters for comparison to the samples from this study (in pie chart format, or similar). The TCGA cohort could similarly be analyzed for primary, untreated (androgen-dependent) prostate cancer.

Major Comment #2: The authors provide some comments regarding the relative abundance of certain subtype-defining lesions (such as TMPRSS2-ETS fusions) among their metastatic CRPC samples, in comparison to prostate cancer as a whole (lines 80-82). Although Figure 2 and Supplementary Figure 3D-3E display the frequencies of SPOP, FOXA1, and IDH1 mutations as well, both of these figures would be of greater interest if they included localized CRPC and/or ADPC controls (generated from publicly available datasets). These comparisons would help place these findings into a larger prostate cancer context and show what is different and unique about metastatic CRPC.

Major Comment #3: The authors state in lines 155-156 that "This study also showed that a large population of mCRPC patients do not fall into an as-of-yet clinically-relevant or biologically-clear genotype...". One unresolved question is the extent to which the 8 clusters of metastatic CRPC identified in this study correspond to known subtypes of prostate cancer. The authors point out in the Introduction section (lines 51-53) that a similar study of primary prostate cancer (reference #4) was able to classify 74% of cases into 7 defined subtypes based on a small number of mutations. If the authors perform similar subtyping of their 197 metastatic CRPC samples, to what extent do they overlap with each of the known subtypes? It would be a potentially interesting way to gain insight into whether certain types of genomic instability play causal roles in the acquisition of key driver mutations, or vice versa.

Major Comment #4: Many of the insights in the manuscript come from the 4 clusters of metastatic CRPC cases that are associated with a recognized underlying source of genomic instability. On the other hand, the 4 novel clusters undergo an extensive exploration of what they have in common at the genomic level (Supplementary Figures 6-10) but with limited or difficult-to-interpret results. For example, Cluster G is associated with translocations, Cluster H with insertions, Cluster C with tandem duplications <100kb, and Cluster E (the largest cluster) with nothing in particular. Despite the authors' deliberate selection of 8 as the optimal number of clusters (Supplementary Figure 7A), there seems to be a lingering possibility that clustering is not optimally representing the underlying biology. The association of Clusters G and H with translocations and insertions, respectively, reveals how each type of structural variation was given an individual weight as a criterion for clustering, rather than being grouped together as a single marker of chromosomal-level instability, etc. This modeled the assumption that each type of structural variation might have a different underlying cause. However, if structural variations were grouped together, it would be interesting to see if variables like point mutation signatures (which did not associate with any cluster in this analysis) might have relatively more weight to identify their own cluster. Did the authors try any alternative set of clustering criteria in which two or more types of structural variation were grouped together as a single marker of an over-arching chromosomal instability phenotype? Could they provide some description of the process by which they identified the clustering criteria described in lines 110-111 as the best option?

Minor Comment #1: Figure 3C would likely be easier to interpret if y-axes for multiple tracks belonging to the same target were scaled to the same maximum values. For example, the untreated and R1881-treated AR ChIP-seq tracks within the AR locus for LNCaP could be shown at the same scale, with the range from 0.0 to 1.2 shown on both y-axes.

Minor Comment #2: The authors state in lines 151-154 that they "show the added value of WGS-

based unbiased clustering" for a clinical application. One minor caveat to point out is that the unbiased clustering approach seems to assume that categories are mutually exclusive. Although it is logical that each tumor would need only one predominant source of genomic instability, some samples might have more than one, resulting in biological characteristics in common with multiple clusters. In lines 130-132, the authors describe a sample harboring (at least one?) pathogenic BRCA2 mutation but classified as an MSI tumor based on the unbiased clustering approach. It is unclear how the patient in this case might respond to PARP inhibitors or whether mismatch repair mutations are present as well, but it does raise the possibility that a shortcoming of unbiased clustering might be its susceptibility to placing samples with mixed or marginal characteristics in one category rather than two si

Reviewer #2 (Remarks to the Author):

The current report by van Dessel et al., conducts whole genome sequencing analysis of fresh frozen biopsies derived from Dutch castration resistant prostate cancer patients with either lymph node or distant metastases (bone, liver). The authors detected four distinct and potentially clinically relevant groups including microsatellite instability, homologous recombination deficiency, tandem duplication phenotypes and chromothripsis events. The results of the study are highly clinically significant. A caveat however, is the striking similarity to a recent report in Cell (PMID: 30033370) that describes similar findings in an independent cohort.

Comments

- Is there any correlation between the identified genotypes and overall survival?
- Has the treatment history of the patient influenced the observed genetic heterogeneity?
- Has only one metastasis biopsy been taken from each patient. If more than one (multifocal bone mets for example or bone mets and soft tissue, i.e. liver) is there similarity between the genetic mutations derived from each each site within the same individual?
- The discussion of the obtained data should be in a separate section and the authors findings put in context of previous publications, clearly defining what is similar and what is distinct. Are possible differences due to ethnic backgrounds etc.

Reviewer #1

Introductory Comment – CHIP-seq data

The study includes other experiments that build on the whole genome sequencing data, particularly CHIP-seq studies that yield some interesting findings such as the validation that certain amplification events observed in patients coincide with genome binding sites for the AR and FOXA1 transcription factors. However, these data are different enough from the title of the manuscript and the theme of demonstrating the clinical utility of whole genome sequencing data that they do not help the study form a single cohesive story. Overall, the study provides a valuable analysis of a large number of metastatic, castration-resistant prostate cancer samples, with subtyping based on somatically acquired genomic changes. Specific comments and suggestions are below.

Response:

Our WGS analysis showed that alterations of the AR-pathway are not subgroup specific. Combining our WGS data with CHIP-seq data shows the potential clinical relevance of recurrent alterations in non-coding regions identified with WGS. The combination of WGS data and CHIP-seq data in patient material, in our view firmly establishes the importance of assessing non-coding regions in prostate cancer. As our manuscript describes the potential clinical impact of WGS based stratification, we feel this analysis does fit within the scope of our manuscript.

Major Comment #1 - Incorporation of primary prostate WGS

The clinical details linked to the new samples seem to be biopsy site, age, and previous treatments of the patients, and do not include prospectively collected data. This limits the study's ability to associate their 8 clusters/subtypes with distinct outcomes, as the most important clinical outcome here is the already-achieved status of these samples as metastatic CRPC. These data might be best used to demonstrate the enrichment of certain subtypes among metastatic CRPC and the enhanced ability of certain subtypes of early prostate cancer to progress to this stage. As a result, it may be important to perform a more thorough analysis of other prostate cancer whole genome sequencing datasets for comparison to this metastatic CRPC group. An example of how the authors have already started to do this is their finding that their metastatic CRPC samples have an average tumor mutation burden in coding regions more than 2-fold higher than primary prostate cancer (lines 64-68).

There are several opportunities for additional informatics analyses to better integrate the new findings with similar studies. For example, one of the strengths of the manuscript (the relatively large number of metastatic CRPC biopsies) might be better exhibited by making more comparisons to publicly available WGS data from localized or androgen-dependent disease. Using the clustering criteria identified in this study (ploidy, mutational burden, and the number and relative frequencies of insertions, deletions, tandem duplications and translocations), can the authors analyze public whole-genome sequencing data to determine how well-represented are each of these 8 clusters in the localized CRPC or androgen-dependent stages of prostate cancer? Supplementary figure 2 contains some comparison of tumor mutation burden in metastatic CRPC samples vs. localized CRPC samples. It would be very valuable if the authors expanded this line of investigation to categorize the localized CRPC samples into clusters for comparison to the samples from this study (in pie chart format, or similar). The TCGA cohort could similarly be analyzed for primary, untreated (androgen-dependent) prostate cancer.

Response:

We extended our study by incorporating a cohort of localized prostate cancer (whole-genome sequencing; $n = 210$), of exclusively Gleason score (GS) 6 - 7, in cooperation with Prof. Dr. Paul Boutros. We've extended the comparison between mCRPC and localized prostate cancer by comparing the overall genome-wide tumour mutational burden, the mutational frequency of the driver genes in mCRPC (dN/dS, GISTIC2 and subtype-specific) as shown in **figure 3**, which now also replaced **supplementary figure 2** (lines 136 - 149).

We also note that the difference in TMB between GS 6-7 primary prostate cancer and mCRPC is even more extreme: 0.71 (GS 6 - 7) vs. 2.7 (mCRPC). The previous primary prostate cancer cohort used in the comparison in former **supplementary figure 2** (PCAWG) contained samples ranging between GS 6 to 10 (T1c - T3b stages) resulting in a higher TMB (1.22) probably due to the more advanced cases.

Furthermore, we've performed the same clustering method using slightly different genome-wide aberrations on both the primary prostate cancer and mCRPC cohorts (**supplementary figure 12**). Unfortunately, large insertions and ploidy were not available for the primary cohort due to different methodology on structural variant calling and copy-number alterations.

We did not discover striking novel primary-only genomic subtypes and observed that the cohorts cluster semi-independently. This could be due to the low mutational burden of primary prostate cancers (GS 6 - 7) or due to differences in methodology. Additionally, no samples with *CDK12* coding mutations, only one to possibly three samples with MSI, and only three samples with *BRCA2* coding mutations were observed in the primary prostate cancer dataset. This could reflect a heterogeneous population of patients (or disease stages) and makes a direct comparison between the frequencies of the major clusters (MSI, BRCAness and CDK12) difficult to interpret.

Furthermore, when comparing the mutational frequencies of the driver (dN/dS and GISTIC2) and subtype-specific genes discovered in the mCRPC cohort, we observed treatment-specific alterations such as *AR* amplifications and mutations due to ADT-resistance but also increased aberrations in known markers associated with disease progression and poor prognosis such as *TP53* coding mutations, *MYC* amplifications, *RYBP* loss and 8p/8q aberrations (**figure 3c-e**; lines **142 - 147**)^{1,2}. Indeed, we did not observe any novel metastatic-only drivers, only an increase in mutations for genes also mutated, to a lesser extent, in primary disease.

The current analysis shows a clear enrichment of poor prognostic genes in the metastatic setting, confirming prior studies. We did not identify novel driver genes in the metastatic prostate cancer setting, however, this may be, in part, due to the use of unmatched data. Moreover, technical differences in the analysis of the metastatic vs local prostate cancer dataset may decrease the sensitivity of this analysis. Thus we provide a broad context for our data however we are unable to exclude potential low frequency driver genes that are specific for the metastatic setting.

Major Comment #2 – Comparison to primary prostate WGS

The authors provide some comments regarding the relative abundance of certain subtype-defining lesions (such as TMPRSS2-ETS fusions) among their metastatic CRPC samples, in comparison to prostate cancer as a whole (lines 80-82). Although Figure 2 and Supplementary Figure 3d-3e display the frequencies of SPOP, FOXA1, and IDH1 mutations as well, both of these figures would be of greater interest if they included localized CRPC and/or ADPC controls (generated from publicly available datasets). These comparisons would help place these findings into a larger prostate cancer context and show what is different and unique about metastatic CRPC.

Response:

We've extended the investigation into the differences in mutational landscape between localized and metastatic disease as discussed in major comment #1 of this reviewer. In addition, please also see the discussion of major comment #3 of this reviewer for additional information on primary disease subtype-specific lesions with a focus on *SPOP*, *FOXA1* and *IDH1* (as defined by the TCGA) in mCRPC.

By comparing to the primary prostate adenocarcinoma cohort (see major comment #1), we could better investigate major differences between both disease settings (**figure 3; supplementary table 3 - Sheet M and N**). We observed that the overall tumour mutational burden increased over time and disease progression; 0.71 in primary disease (GS 6-7) versus 2.7 in mCRPC (line **145; figure 3a**). This increase in mutational burden over time and disease progression was also observed for structural variants (**figure 3b**). This change in mutational burden could also be seen when both cohorts are analysed together, using unsupervised hierarchical clustering or PCA on shared and basic whole-genome characteristics, which revealed a separation of both cohorts primarily on mutational burden (**supplementary figure 12; lines 231 - 238**).

We feel these additions more clearly place our data in the larger context of prostate cancer genomics and improve the quality of the manuscript.

Major Comment #3 - Subtyping based on mutually-exclusive aberrations

The authors state in lines 155-156 that “This study also showed that a large population of mCRPC patients do not fall into an as-of-yet clinically-relevant or biologically-clear genotype...”. One unresolved question is the extent to which the 8 clusters of metastatic CRPC identified in this study correspond to known subtypes of prostate cancer. The authors point out in the Introduction section (lines 51-53) that a similar study of primary prostate cancer (reference #4) was able to classify 74% of cases into 7 defined subtypes based on a small number of mutations. If the authors perform similar subtyping of their 197 metastatic CRPC samples, to what extent do they overlap with each of the known subtypes? It would be a potentially interesting way to gain insight into whether certain types of genomic instability play causal roles in the acquisition of key driver mutations, or vice versa.

Response:

We agree with this insightful remark to better incorporate and highlight the previous mutational clustering of localized prostate cancer as shown by the TCGA⁷ based on *SPOP*, *FOXA1* and *IDH1* mutations and copy-number aberrations and ETS-family gene fusions (and overexpression) per promiscuous partner (*ERG*, *ETV1*, *ETV4* and *FLI1*). Half of *ETV1*, *ETV4* and *FLI1*-clustered samples in the TCGA paper were based on overexpression. Unfortunately, we currently lack matched mRNA-sequencing and therefore cannot observe overexpression of fused ETS-family members, restricting us to only characterize the genomic breaks of these promiscuous partners.

We’ve included an additional supplementary figure (**Supplementary figure 10**; lines **217 - 229**) showing the presence of these cluster-specific genomic mutations and genomic fusions in our CPCT-02 metastatic mCRPC (WGS) cohort. To reduce differences between the two studies, we only incorporated the mutational categories as used by TCGA and comparable to ours; coding mutations, deep deletions and presence of fusion, ignoring deep gains.

When comparing the frequencies of these cluster-specific aberrations (with and without overexpression status) between the localized prostate cancers from the TCGA paper versus our mCRPC cohort (**supplementary figure 10b**), we observe that all cluster-specific mutations, except *FOXA1* mutations and *ETV1* fusions, share similar mutational frequencies. The lower number of ETS-fusions, compared to TCGA, could possibly be attributed to missing matched RNA sequencing which can be more sensitive for fusion-detection, when compared to detecting genomic breaks using WGS, especially for highly-expressed fusions. Interestingly, we do not observe full mutual-exclusivity between *ERG*-fusion and *FOXA1*, *IDH1*, and *SPOP* mutations nor do we detect *IDH1*-mutant patients without an accompanying cluster-specific mutation.

From this, we conclude that whole-genome sequencing of the metastatic samples reveals genomic landscapes complementing mutually-exclusive genomic mutations and partially superseding such observed mutations, such as samples without observed pathogenic *CDK12* mutation or *BRCA2* mutation falling within our *CDK12* and *BRCA2* cluster (cluster B and D, respectively; lines **249 - 264**)

Using this mutation-driven TCGA clustering, we cannot differentiate patients into MSI (WGS cluster A), *CDK12* (WGS cluster B) and *BRCA2* (WGS cluster D) subgroups as these are spread throughout the TCGA-driven clusters. The resulting ETS-based fusion, *SPOP* and *FOXA1* clusters do not harbor specific genomic landscapes linked to a single species of genomic instability or genotype.

In addition, we could not derive further statistical enrichment or depletion ($q \leq 0.05$; two-sided Fisher’s Exact Test with Benjamini-Hochberg corrections) between mutation-driven clusters and additional mutations, kataegis and chromothripsis, except the known enrichment of homozygous *CHD1* deletions in the *SPOP*-cluster (lines **227 - 229**).

Major Comment #4 - Re-evaluation of clustering criteria

Many of the insights in the manuscript come from the 4 clusters of metastatic CRPC cases that are associated with a recognized underlying source of genomic instability. On the other hand, the 4 novel clusters undergo an extensive exploration of what they have in common at the genomic level (Supplementary Figures 6-10) but with limited or difficult-to-interpret results. For example, Cluster G is associated with translocations, Cluster H with insertions, Cluster C with tandem duplications <100kb, and Cluster E (the largest cluster) with nothing in particular. Despite the authors' deliberate selection of 8 as the optimal number of clusters (Supplementary Figure 7a), there seems to be a lingering possibility that clustering is not optimally representing the underlying biology. The association of Clusters G and H with translocations and insertions, respectively, reveals how each type of structural variation was given an individual weight as a criterion for clustering, rather than being grouped together as a single marker of chromosomal-level instability, etc. This modeled the assumption that each type of structural variation might have a different underlying cause. However, if structural variations were grouped together, it would be interesting to see if variables like point mutation signatures (which did not associate with any cluster in this analysis) might have relatively more weight to identify their own cluster. Did the authors try any alternative set of clustering criteria in which two or more types of structural variation were grouped together as a single marker of an over-arching chromosomal instability phenotype? Could they provide some description of the process by which they identified the clustering criteria described in lines 110-111 as the best option?

Response:

The reviewer notes an interesting observation which we investigated further. We performed an additional PCA to identify possible combinations of the proposed clustering metrics (TMB, mean genome ploidy, relative percentage of SV categories and total SV). This PCA (**Supplementary figure 6a-c**) indeed revealed two principal components (PC 3 and PC 7) with a possible link between SV categories, specifically insertions and translocations, as main contributors of their respective components and as noted by the reviewer.

We combined these two categories (insertions and translocations) into a single metric summarizing the relative frequency of both categories, and removed them as individual features, in an additional clustering analysis using the same methodology (Ward.D method on 1-pearson correlation) (**review figure 1**).

We then cut this clustering outcome into a maximum of 5 to 15 clusters (optimum according to Elbow's method around 7-9) and tested each of these cut-offs for a correlation or enrichment with a two-sided Fisher's Exact Test (BH) to known onco-genes and tumor suppressors, dN/dS and GISTIC2 enriched genes, kataegis, chromothripsis and ETS fusions. The bottom track of **review figure 1** shows the cluster outcomes of the 'original' clustering as shown in the manuscript.

As shown in the bottom tracks, the major subgroups as described in the manuscript (MSI, BRCAness, CDK12 and chromothripsis) were observed without major deviations and only cluster E and F (righthand-side of figure) now show a small mixture and a distinct new cluster of several cluster F samples (chromothripsis) based on the total number of SV. No new clusters (ranging from max. 5 to 15 clusters) could be attributed to another new type of aberration or mutant gene. It would seem that the samples with highest genome instability, most aberrant ploidy, SV and certain combinations of SV are indeed the samples with chromothripsis.

We've also added an additional figure (**supplementary figure 6d-j**) highlighting, partially, the rationale behind the current criteria for the unsupervised clustering and discussed the potential pitfalls of our clustering analysis (lines **282 - 287**).

In this manuscript, we wished to highlight the potential of using genome-wide effects of various genetic drivers such as *CDK12* (tandem duplications), DDR-deficiency (high TMB / InDels), *BRCA1* (small tandem-duplications), *BRCA2* (large deletions) to stratify patients for potential treatments on these effects.

For instance, patients with variants-of-uncertain-significance (VUS) in *BRCA2* could potentially benefit from PARP-inhibitors if on genome-wide level the genomic scars (or mutational signatures) associated with pathogenic *BRCA2* variants are found.

Future research into identifying clinically-actionable mechanisms will potentially yield new distinct genomic (and mutational) features which can then be incorporated into an overall classifier to better stratify patients. This manuscript shows that this is already currently possible using this approach as a proof-of-concept and shows that this type of stratification will likely only get improved as new features are discovered and more advanced stratification approaches are employed (such as machine learning or 'classical' classifier approaches).

Minor Comment #1 - Y-axis Figure 3c (ChIP-Seq)

Figure 3c would likely be easier to interpret if y-axes for multiple tracks belonging to the same target were scaled to the same maximum values. For example, the untreated and R1881-treated AR ChIP-seq tracks within the AR locus for LNCaP could be shown at the same scale, with the range from 0.0 to 1.2 shown on both y-axes.

Response:

We have scaled the y-axis to the same maximum values for tracks belonging to the same target (AR - LNCaP + AR LNCaP R1881 and AR VCaP + AR VCaP Bicalutamide Resistant per shown loci). Indeed, this provides a better visual distinction between both conditions (**Figure 4c**).

Minor Comment #2 - Multiclass patients

The authors state in lines 151-154 that they “show the added value of WGS-based unbiased clustering” for a clinical application. One minor caveat to point out is that the unbiased clustering approach seems to assume that categories are mutually exclusive. Although it is logical that each tumor would need only one predominant source of genomic instability, some samples might have more than one, resulting in biological characteristics in common with multiple clusters. In lines 130-132, the authors describe a sample harboring (at least one?) pathogenic BRCA2 mutation but classified as an MSI tumor based on the unbiased clustering approach. It is unclear how the patient in this case might respond to PARP inhibitors or whether mismatch repair mutations are present as well, but it does raise the possibility that a shortcoming of unbiased clustering might be its susceptibility to placing samples with mixed or marginal characteristics in one category rather than two si

Response:

With the current approach, we show that the general mCRPC population can be categorized into distinct genotypes using whole-genome characteristics but, for each distinct patient, additional information can be extracted such as additional aberrations suitable for potential therapy choices.

The reviewer raises a valid pitfall of the employed hierarchical clustering scheme and we are currently extending these findings into a separate classifier, which upon completion, can classify patients into one or multiple distinct subtypes based on the genomic characteristics and mutations as shown within this manuscript. The single MSI-tumor, which also displays a pathogenic *BRCA2* mutation, is indeed only clustered into a single cluster (Cluster A - MSI) based on its predominant genotype. This sample displays an increased number of structural variants, ploidy associated with the *BRCA2* samples (cluster D), however it is missing the characteristic number of genomic deletions (<100kb) and BRCA mutational signature whilst displaying the MSI signature and mutations in MMR-related genes (*MLH1*, *POLE*, *POLD3* and *BLM*).

In conclusion, with an extended classifier a patient-specific profile can be established which encompasses one or more genotypes whilst also considering specific targetable mutations. The clusters on the left-hand side (A – D) all have rather-defining genomic characteristics suggesting distinct routes of early somatic evolution based on the afflicted driver-gene whilst the right-hand side (E – F) could indeed benefit from an approach which utilizes multi-classing as described above. We agree with the comment of the reviewer and have adapted the discussion to emphasize this limitation with our chosen hierarchical clustering approach (lines **287 - 294**).

Reviewer #2

Major Comment #1 - Correlation to survival

Is there any correlation between the identified genotypes and overall survival?

Response:

The CPCT-02 study on which we are reporting collected biopsies of metastatic prostate cancer patients irrespective of the timing with respect to their stage of the disease. This has led to a broad heterogeneity of number and type of prior treatments (see response to Major comment #2 of this reviewer). Therefore, the prognosis of patients differs greatly depending on the treatment history, as the treatments for mCRPC patients nowadays significantly impacts overall survival⁸⁻¹⁴.

However, one may argue that the genomic data for each patient is relatively stable and therefore survival analysis based on the classification into subgroups may still be reasonable. In our opinion there is ample evidence that the genomic data derived from patient alters over time and some well-defined prognostic factors such as p53 mutations, Rb1 loss and AR aberrations will change over time^{15,16}. Therefore, we do not believe that reporting survival data in the current manuscript allows for an improved understanding without correcting for all possible biases or setting up a more homogeneous patient cohort. These biases will be comprised of genomic features that have changed over time (as illustrated in response to 'Major Comment #1' from Reviewer #1). The subgroup determination may represent a stable factor; however, this is subject to future studies in which we will try to obtain biopsies from the same patient at multiple time points to prove the subgroup classification is stable over the course of the disease. Moreover, the clinical features that potentially affect prognosis such as whether the patient had initially high-risk metastatic disease or had progressed years after local treatment would need to be incorporated and is currently not (yet) available for this cohort. Treatment response factors such as the time on treatment for the first ADT treatment, disease load as measured by alkaline phosphatase, lactate hydrogenase levels and other general prognostic factors or response to later lines of therapy may also affect prognosis. The description of all of these clinical features is beyond the scope of this manuscript and therefore we feel reporting survival data without the proper context will not add value to this manuscript and would represent an incomplete view on all these varying factors.

We did incorporate a section in the discussion to reflect that the lack of homogeneity of the population makes analyses on the prognostic value of these data inappropriate (lines 296 - 302).

Major Comment #2 – Treatment history and genetic heterogeneity

Has the treatment history of the patient influenced the observed genetic heterogeneity?

Response:

In this manuscript, the treatment history (number, sequence and type of previous therapies) differs significantly between patients. Therefore, it is not feasible to perform robust statistical analysis on the correlation between genomic data and every individual pre-treatment.

However, to still provide some insight into a possible correlation, we have grouped diverse pre-treatments into ten general treatment groups:

- Only secondary anti-hormonal therapy (with / without radio-nucleotides);
- Only chemo-treatment (with / without radionucleotide);
- Only radionucleotides;
- Secondary anti-hormonal therapy + one chemo-treatment (with / without radio-nucleotides);
- Secondary anti-hormonal therapy + two chemo-treatment (with / without radio-nucleotides);
- No additional treatment after ADT.

After splitting the mCRPC cohort into these 10 groups based on generalized pre-treatment history, we performed a statistical analysis using a two-sided Fisher's Exact Test with Benjamini-Hochberg corrections. We could not detect statistical enrichment or depletion ($q \leq 0.05$) between these supervised treatment groups and our defined genomic clusters (cluster A - H), nor did we observe an enrichment or depletion of mutations, MSI status, kataegis and chromothripsis in relation to treatment group.

We have added these treatment groups as a separate track below an additional figure **reviewFig2_OncoplotWithPretreatment.svg** to also visualize that no striking correlation between (general) treatment and genetic heterogeneity can be seen. This could potentially also be in relation to the heterogeneous nature of the current cohort as described in Major Comment #1 from this reviewer.

We've extended the results section with a paragraph on this (lines **150 - 154**) and added an additional supplemental table 3 to show the results from this analysis.

Major Comment #3 – Multiple biopsies

Has only one metastasis biopsy been taken from each patient? If more than one (multifocal bone mets for example or bone mets and soft tissue, i.e. liver) is there similarity between the genetic mutations derived from each site within the same individual?

Response:

Due to the nature of our clinical protocol, we were only able to obtain one biopsy per patient at a specific time point from a single lesion. For some patients, we have collected multiple biopsies over time and lesion, however in this analysis we included only the first biopsy sample taken.

The number of paired biopsies is relatively low compared to the entire dataset and was only taken after progression on a systemic treatment. These biopsies are intended for the discovery of treatment resistance associated aberrations and will thus not answer the intra-patient heterogeneity question.

Nevertheless, we agree the reviewer raises an interesting question we are unfortunately unable to answer using this dataset.

Major Comment #4 - Reformatting and ethnical differences

The discussion of the obtained data should be in a separate section and the authors findings put in context of previous publications, clearly defining what is similar and what is distinct. Are possible differences due to ethnic backgrounds etc.

Response:

We have adjusted the format of our manuscript to the format of Nature Communications and included the sections *Introduction*, *Results* and *Discussion*. In the Results and Discussion sections we discuss our findings in light of current literature (lines 117 - 121; 122 - 124; 128 - 131; 192 - 195; 203 - 205; 208 - 210; 266 - 271).

We did not collect data on ethnic background in our study and this can therefore not be related to any results. It is known race and ethnicity are contributory to prostate cancer risk, aggressiveness, and prognosis. In general, we have a predominant Caucasian population in the Netherlands. As our cohort focusses on castration-resistant prostate cancer, there is already a selection bias for aggressive prostate tumors thus race and/or ethnicity may have less impact on the genomic characteristics. Nevertheless, the lack of correlation of our genomic data with clinical characteristics such as ethnicity is a limitation of our study; we have addressed this in the discussion.

References

1. Ulz, P. *et al.* Whole-genome plasma sequencing reveals focal amplifications as a driving force in metastatic prostate cancer. *Nature Communications* **7**(2016).
2. Robinson, D. *et al.* Integrative Clinical Genomics of Advanced Prostate Cancer. *Cell* **162**, 454 (2015).
3. Chan, T.A. *et al.* Development of tumor mutation burden as an immunotherapy biomarker: utility for the oncology clinic. *Annals of Oncology* **30**, 44-56 (2019).
4. Zehir, A. *et al.* Mutational landscape of metastatic cancer revealed from prospective clinical sequencing of 10,000 patients. *Nature Medicine* **23**, 703-+ (2017).
5. Samstein, R.M. *et al.* Tumor mutational load predicts survival after immunotherapy across multiple cancer types. *Nature Genetics* **51**, 202-+ (2019).
6. Lui, G.Y.L., Grandori, C. & Kemp, C.J. CDK12: an emerging therapeutic target for cancer. *Journal of Clinical Pathology* **71**, 957-962 (2018).
7. Cancer Genome Atlas Research, N. The Molecular Taxonomy of Primary Prostate Cancer. *Cell* **163**, 1011-25 (2015).
8. Kantoff, P.W. *et al.* Sipuleucel-T immunotherapy for castration-resistant prostate cancer. *N Engl J Med* **363**, 411-22 (2010).
9. Beer, T.M. *et al.* Enzalutamide in Metastatic Prostate Cancer before Chemotherapy. *New England Journal of Medicine* **371**, 424-433 (2014).
10. Bono, J.S., Oudard, S. & Ozguroglu, M. Prednisone plus cabazitaxel or mitoxantrone for metastatic castration-resistant prostate cancer progressing after docetaxel treatment: a randomised open-label trial (vol 376, pg 1147, 2010). *Lancet* **377**, 720-720 (2011).
11. de Bono, J.S. *et al.* Abiraterone and increased survival in metastatic prostate cancer. *N Engl J Med* **364**, 1995-2005 (2011).
12. Ryan, C.J. *et al.* Final Overall Survival (Os) Analysis of Cou-Aa-302, a Randomized Phase 3 Study of Abiraterone Acetate (Aa) in Metastatic Castration-Resistant Prostate Cancer (Mcrpc) Patients (Pts) without Prior Chemotherapy. *Annals of Oncology* **25**(2014).
13. Scher, H.I. *et al.* Increased Survival with Enzalutamide in Prostate Cancer after Chemotherapy. *New England Journal of Medicine* **367**, 1187-1197 (2012).
14. Parker, C. *et al.* Alpha Emitter Radium-223 and Survival in Metastatic Prostate Cancer. *New England Journal of Medicine* **369**, 213-223 (2013).
15. Hamid, A.A. *et al.* Compound Genomic Alterations of TP53, PTEN, and RB1 Tumor Suppressors in Localized and Metastatic Prostate Cancer. *Eur Urol* **76**, 89-97 (2019).
16. Abida, W. *et al.* Prospective Genomic Profiling of Prostate Cancer Across Disease States Reveals Germline and Somatic Alterations That May Affect Clinical Decision Making. *Jco Precision Oncology* **1**(2017).

Summary of major revisions

- Figure 1
 - Changed figure 1b to a more aesthetically-pleasing figure, no change to data.
- Figure 2
 - Changes colors of the biopsy location track to correspond to the colors in figure 1b.
- Figure 3
 - Included a new main figure highlighting the main mutational differences between localized and metastatic prostate cancer.
- Figure 4c
 - Scaled the y-axis to the same maximum values for tracks belonging to the same target (AR - LNCaP + AR LNCaP R1881 and AR VCaP + AR VCaP Bicalutamide Resistant per shown loci).
- Figure 5
 - Changes colors of the biopsy location track to correspond to the colors in figure 1b.
- Supplementary table 3
 - Changed CPCT02 identifiers to global HMF identifiers in line with the pan-cancer release of the dataset.
 - Added clinical metadata of the WGS samples (Sheet A) and renamed subsequent sheets accordingly.
- Supplementary figure 6 - Added a new figure to highlight the distributions of SV and employed genomic features in clustering.
- Suppl. figure 10 – Clustering on TCGA scheme.
 - Added new figure showing the TCGA-based clustering metrics on our mCRPC cohort.
- Suppl. Figure 12 – Unsupervised clustering of mCRPC and primary prostate cancer.
 - Added new showing unsupervised clustering of mCRPC and primary prostate cancer to highlight disease-specific clusters.
- Removed former supplementary figure 2 on TMB between primary and metastatic disease as this is now incorporated in figure 6.
- Manuscript:
 - Changed figure numbers of main figures and supplementary figures, as suppl. figure 2 was integrated into main figure 6 and a new main figure was added.
 - Authors
 - Added additional co-authors based on work after initial manuscript (comparison localized prostate cancer)
 - Abstract:
 - Added findings on primary cohort to the abstract.
 - Main text was redrafted to match the standard format of Nature Communications (*Introduction, Results, Discussion*).

- Due to the extended word count additional text was added to the *Introduction*, to clarify figures and to incorporate reviewer comments.
- Results
 - Subheadings were added.
 - Line 130-135: Switched driver gene presentation, first amplification/deletions using GISTIC2 and then those detected by dn/ds.
 - Line 140-142: Added mention of novel mutational signature analysis.
 - Line 157-161: Added information on the analysis of pretreatment.
 - Line 185-190: Extended details of clustering.
 - Line 224-236: Added TCGA-based clustering .
 - Line 238-245: Added paragraph to accommodate the results of the clustering comparison to primary prostate cancer .
- Discussion
 - Added and extended discussion based on current findings and analysis.
 - Added paragraph on limitations of our study.
 - Added paragraph discussing CPCT-02 study in more detail.
- Methodology
 - Added M&M on the clustering on TCGA criteria and acquisition of the data.
 - Added M&M on the Comparison of the mutational landscape between unmatched primary prostate cancer and mCRPC.
 - Removed M&M on the comparison of the TMB from PCAWG cohort to mCRPC.
 - Added M&M on the correlation of pre-treatment history with detected aberrations and clusters.
- Figure legends
 - Supplementary table 3 - Added new sheets to the legends and corrected sheet numbers.
 - Added figure legends for new figures (figure 6 and suppl. figure 10, 11 and 12).

REVIEWERS' COMMENTS:

Reviewer #1 (Remarks to the Author):

In the authors' response letter and revised documents, they have made a sincere effort to address the reviewer comments. This work has provided satisfactory answers to some of the comments (Reviewer 1 Major Comments 1, 3 and 4, Reviewer 1 Minor Comment 2, and Reviewer 2 Major Comments 1, 2, 3 and 4) and partial answers to others (Reviewer 1 Introductory Comment, Reviewer 1 Major Comment 2, Reviewer 1 Minor Comment 1). Yet some of the comments point out significant limitations of the current version of the study. For example, while the authors have certainly improved their study of metastatic CRPC by adding comparisons to localized ADPC data, the dramatic differences between them make it clear that a better understanding of metastatic CRPC would be facilitated instead by a comparison to localized CRPC that is not present here. The original experimental design (analyzing metastatic CRPC samples only) was never designed to facilitate association of the identified subtypes/clusters with clinical outcomes, and the newly added analysis of localized ADPC has not detected the relevant subtypes/clusters sufficiently in those samples to enable any insights into the relative contributions of the genomic instability types to clinical outcomes. Thus, the absence of a relationship of subtypes to outcomes limits insight into their clinical utility at this point. In addition, the ChIP-seq data are overemphasized at points in the manuscript, in that the experiment is not well-controlled enough or complemented by enough additional data or integration with the literature to enable any novel conclusions that add to the field. Nevertheless, a significant amount of work has been added to make the study more thorough and more integratable with similar published studies. Below are additional comments on some of the authors' responses and edits within the figures and documents.

Reviewer #1 – Introductory Comment – ChIP-seq data

The study includes other experiments that build on the whole genome sequencing data, particularly ChIP-seq studies that yield some interesting findings such as the validation that certain amplification events observed in patients coincide with genome binding sites for the AR and FOXA1 transcription factors. However, these data are different enough from the title of the manuscript and the theme of demonstrating the clinical utility of whole genome sequencing data that they do not help the study form a single cohesive story. Overall, the study provides a valuable analysis of a large number of metastatic, castration-resistant prostate cancer samples, with subtyping based on somatically acquired genomic changes. Specific comments and suggestions are below.

Response:

Our WGS analysis showed that alterations of the AR-pathway are not subgroup specific. Combining our WGS data with ChIP-seq data shows the potential clinical relevance of recurrent alterations in non-coding regions identified with WGS. The combination of WGS data and ChIP-seq data in patient material, in our view firmly establishes the importance of assessing non-coding regions in prostate cancer. As our manuscript describes the potential clinical impact of WGS based stratification, we feel this analysis does fit within the scope of our manuscript.

Reviewer New Comment:

While it is understandable that the authors feel that the ChIP-seq data in Figure 4C are relevant to the study and worth including among the data, it remains difficult to make strong conclusions from them, other than to confirm that some AR, FOXA1, and H3K27ac peaks detectable in cell line models are also present in metastatic patient tissues. We question whether the conclusion that "From WGS and ChIP-seq data, we show that regulators in non-coding regions of MYC and AR are transcriptionally active and highlight the central role of AR signaling in tumor progression" is worth mentioning in the Abstract. On the other hand, the authors' response makes a good point about "the potential clinical relevance of recurrent alterations in non-coding regions identified with WGS". Perhaps this is the more Abstract-worthy finding facilitated by the ChIP-seq data? Some

shortcomings in the ChIP-seq design of Figure 4C are described below, along with some resulting caveats.

The authors focus on 2 amplifications (at an enhancer near the AR locus and at 8q24.21) observed in the 197-patient WGS dataset and examine those 2 loci by ChIP-seq generated from 2 patients (A and B). Patients A and B were chosen because they exhibited the AR enhancer amplification of interest, but the experiment did not include control tissues that did not exhibit this amplification, either from other metastatic or non-metastatic patients lacking the amplification, or from non-tumor tissue. In fact, the matching peaks in the stimulated LNCaP (without AR amplification) and VCaP cells (with AR amplification) indicate that the presence of AR, FOXA1, or H3K27 signal at this enhancer is not unique to prostate cancer cells with AR enhancer amplification. The reader can only conclude that the amplification event coincides with an enhancer region that is marked by AR, FOXA1, or H3K27ac signal. The reader's confidence that the amplification is driving the amount of AR, FOXA1, or H3K27 signal is highly dependent on existing literature about AR enhancer amplifications. The authors have generated ChIP-seq data from Patients A and B that are consistent with the literature and with the cell line controls, but are not well-controlled enough or sufficiently complemented by RNA-level data to enable stand-alone conclusions. The discussion of Figure 4C should be expanded to mention some of these caveats.

The presentation of results from 8q24.21 (also in Figure 4C) is slightly problematic too, in the sense that the Abstract mentions MYC. One of the challenges of interpreting ChIP-seq data is the difficulty of confidently assigning signal to a particular gene based on proximity alone, unless complementary RNA level data is presented showing that the gene expression is dependent on the presence/absence of the ChIP target. Without more direct evidence that PCAT1 or MYC expression is dependent on the presence of AR or FOXA1, it would not be possible to confidently conclude that the observed peaks control the expression of either gene. One way to strengthen this could be to compare the proposed MYC peak location with at least one other study reporting that AR binds to the MYC locus to control its expression (Gao L, Schwartzman J, Gibbs A, Lisac R, Kleinschmidt R, Wilmot B, Bottomly D, Coleman I, Nelson P, McWeeney S, Alumkal J. Androgen receptor promotes ligand-independent prostate cancer progression through c-Myc upregulation. *PLoS One*. 2013 May 21;8(5):e63563.). Possibly the authors could determine whether the amplified region coincides with a MYC-regulating AR binding site characterized in that study? This would strengthen the Abstract's currently too-strong attribution of this amplified region to the control of the MYC locus.

Reviewer 1 Major Comment #1 - Incorporation of primary prostate WGS

The clinical details linked to the new samples seem to be biopsy site, age, and previous treatments of the patients, and do not include prospectively collected data. This limits the study's ability to associate their 8 clusters/subtypes with distinct outcomes, as the most important clinical outcome here is the already-achieved status of these samples as metastatic CRPC. These data might be best used to demonstrate the enrichment of certain subtypes among metastatic CRPC and the enhanced ability of certain subtypes of early prostate cancer to progress to this stage. As a result, it may be important to perform a more thorough analysis of other prostate cancer whole genome sequencing datasets for comparison to this metastatic CRPC group. An example of how the authors have already started to do this is their finding that their metastatic CRPC samples have an average tumor mutation burden in coding regions more than 2-fold higher than primary prostate cancer (lines 64-68). There are several opportunities for additional informatics analyses to better integrate the new findings with similar studies. For example, one of the strengths of the manuscript (the relatively large number of metastatic CRPC biopsies) might be better exhibited by making more comparisons to publicly available WGS data from localized or androgen-dependent disease. Using the clustering criteria identified in this study (ploidy, mutational burden, and the number and relative frequencies of insertions, deletions, tandem duplications and translocations), can the authors analyze public whole-genome sequencing data to determine how well-represented are each of these 8 clusters in the localized CRPC or androgen-dependent stages of prostate

cancer? Supplementary figure 2 contains some comparison of tumor mutation burden in metastatic CRPC samples vs. localized CRPC samples. It would be very valuable if the authors expanded this line of investigation to categorize the localized CRPC samples into clusters for comparison to the samples from this study (in pie chart format, or similar). The TCGA cohort could similarly be analyzed for primary, untreated (androgen-dependent) prostate cancer.

Response:

We extended our study by incorporating a cohort of localized prostate cancer (whole-genome sequencing; n = 210), of exclusively Gleason score (GS) 6 - 7, in cooperation with Prof. Dr. Paul Boutros. We've extended the comparison between mCRPC and localized prostate cancer by comparing the overall genome-wide tumour mutational burden, the mutational frequency of the driver genes in mCRPC (dN/dS, GISTIC2 and subtype-specific) as shown in figure 3, which now also replaced supplementary figure 2 (lines 136 - 149).

We also note that the difference in TMB between GS 6-7 primary prostate cancer and mCRPC is even more extreme: 0.71 (GS 6 - 7) vs. 2.7 (mCRPC). The previous primary prostate cancer cohort used in the comparison in former supplementary figure 2 (PCAWG) contained samples ranging between GS 6 to 10 (T1c - T3b stages) resulting in a higher TMB (1.22) probably due to the more advanced cases.

Furthermore, we've performed the same clustering method using slightly different genome-wide aberrations on both the primary prostate cancer and mCRPC cohorts (supplementary figure 12). Unfortunately, large insertions and ploidy were not available for the primary cohort due to different methodology on structural variant calling and copy-number alterations.

We did not discover striking novel primary-only genomic subtypes and observed that the cohorts cluster semi-independently. This could be due to the low mutational burden of primary prostate cancers (GS 6 - 7) or due to differences in methodology. Additionally, no samples with CDK12 coding mutations, only one to possibly three samples with MSI, and only three samples with BRCA2 coding mutations were observed in the primary prostate cancer dataset. This could reflect a heterogeneous population of patients (or disease stages) and makes a direct comparison between the frequencies of the major clusters (MSI, BRCAness and CDK12) difficult to interpret.

Furthermore, when comparing the mutational frequencies of the driver (dN/dS and GISTIC2) and subtype-specific genes discovered in the mCRPC cohort, we observed treatment-specific alterations such as AR amplifications and mutations due to ADT-resistance but also increased aberrations in known markers associated with disease progression and poor prognosis such as TP53 coding mutations, MYC amplifications, RYBP loss and 8p/8q aberrations (figure 3c-e; lines 142 - 147)^{1,2}. Indeed, we did not observe any novel metastatic-only drivers, only an increase in mutations for genes also mutated, to a lesser extent, in primary disease.

The current analysis shows a clear enrichment of poor prognostic genes in the metastatic setting, confirming prior studies. We did not identify novel driver genes in the metastatic prostate cancer setting, however, this may be, in part, due to the use of unmatched data. Moreover, technical differences in the analysis of the metastatic vs local prostate cancer dataset may decrease the sensitivity of this analysis. Thus we provide a broad context for our data however we are unable to exclude potential low frequency driver genes that are specific for the metastatic setting.

Reviewer #2 Major Comment #1 - Correlation to survival

Is there any correlation between the identified genotypes and overall survival?

Response:

The CPCT-02 study on which we are reporting collected biopsies of metastatic prostate cancer patients irrespective of the timing with respect to their stage of the disease. This has led to a

broad heterogeneity of number and type of prior treatments (see response to Major comment #2 of this reviewer). Therefore, the prognosis of patients differs greatly depending on the treatment history, as the treatments for mCRPC patients nowadays significantly impacts overall survival⁸⁻¹⁴.

However, one may argue that the genomic data for each patient is relatively stable and therefore survival analysis based on the classification into subgroups may still be reasonable. In our opinion there is ample evidence that the genomic data derived from patient alters over time and some well-defined prognostic factors such as p53 mutations, Rb1 loss and AR aberrations will change over time^{15,16}. Therefore, we do not believe that reporting survival data in the current manuscript allows for an improved understanding without correcting for all possible biases or setting up a more homogeneous patient cohort. These biases will be comprised of genomic features that have changed over time (as illustrated in response to 'Major Comment #1' from Reviewer #1). The subgroup determination may represent a stable factor; however, this is subject to future studies in which we will try to obtain biopsies from the same patient at multiple time points to prove the subgroup classification is stable over the course of the disease. Moreover, the clinical features that potentially affect prognosis such as whether the patient had initially high-risk metastatic disease or had progressed years after local treatment would need to be incorporated and is currently not (yet) available for this cohort. Treatment response factors such as the time on treatment for the first ADT treatment, disease load as measured by alkaline phosphatase, lactate hydrogenase levels and other general prognostic factors or response to later lines of therapy may also affect prognosis. The description of all of these clinical features is beyond the scope of this manuscript and therefore we feel reporting survival data without the proper context will not add value to this manuscript and would represent an incomplete view on all these varying factors.

We did incorporate a section in the discussion to reflect that the lack of homogeneity of the population makes analyses on the prognostic value of these data inappropriate (lines 296 - 302).

Reviewer New Comment:

The original Reviewer 1 Comment 1 was proposed as a way to gain insight into clinical outcomes associated with each of the 8 subtypes identified by the authors, through estimation of the proportion of each subtype found in earlier-stage tumors, and determination of their relative enrichment or depletion from the metastatic tumors. From the authors' response to that comment and to the related Reviewer 2 Comment 2, it seems that this was performed satisfactorily but that the reviewer's underlying question was misguided or impossible to evaluate in this manner, in that these subtypes were either present in the primary prostate cancer dataset at much reduced frequencies and/or were difficult to detect (no samples with CDK12 coding mutations, few with MSI or coding mutations in BRCA2) due to low TMB, etc. Perhaps a non-metastatic CRPC cohort would be more appropriate for the proposed comparison. But this would be outside the scope of the present study, and the additional analyses are satisfactory. Nevertheless, as Reviewer 2 Comment 1 also points out, the inability to link any of the 8 clusters/subtypes from this study to clinical outcome relative to the others remains at least a minor shortcoming. The literature suggests that the type of genomic instability is potentially clinically meaningful, at least as a predictor of a sustained response to immune checkpoint blockade. However, in the present study, the comparison to primary tumors instead has indicated simply that tumor mutation burden, regardless of origin, is strongly associated with progression to metastatic CRPC.

Reviewer 1 Major Comment #2 – Comparison to primary prostate WGS

The authors provide some comments regarding the relative abundance of certain subtype-defining lesions (such as TMPRSS2-ETS fusions) among their metastatic CRPC samples, in comparison to prostate cancer as a whole (lines 80-82). Although Figure 2 and Supplementary Figure 3d-3e

display the frequencies of SPOP, FOXA1, and IDH1 mutations as well, both of these figures would be of greater interest if they included localized CRPC and/or ADPC controls (generated from publicly available datasets). These comparisons would help place these findings into a larger prostate cancer context and show what is different and unique about metastatic CRPC.

Response:

We've extended the investigation into the differences in mutational landscape between localized and metastatic disease as discussed in major comment #1 of this reviewer. In addition, please also see the discussion of major comment #3 of this reviewer for additional information on primary disease subtype-specific lesions with a focus on SPOP, FOXA1 and IDH1 (as defined by the TCGA) in mCRPC.

By comparing to the primary prostate adenocarcinoma cohort (see major comment #1), we could better investigate major differences between both disease settings (figure 3; supplementary table 3 - Sheet M and N). We observed that the overall tumour mutational burden increased over time and disease progression; 0.71 in primary disease (GS 6-7) versus 2.7 in mCRPC (line 145; figure 3a). This increase in mutational burden over time and disease progression was also observed for structural variants (figure 3b). This change in mutational burden could also be seen when both cohorts are analysed together, using unsupervised hierarchical clustering or PCA on shared and basic whole-genome characteristics, which revealed a separation of both cohorts primarily on mutational burden (supplementary figure 12; lines 231 - 238). We feel these additions more clearly place our data in the larger context of prostate cancer genomics and improve the quality of the manuscript.

Reviewer New Comment:

In retrospect, after viewing the dramatic differences between the localized primary (ADPC) cases and metastatic CRPC cases shown in Figure 3, the more informative comparison for better understanding what is unique about metastasis would have been localized CRPC. Nevertheless, the authors are right that they have made strong improvements by highlighting the separation in tumor mutation burden between ADPC and metastatic CRPC and the increased frequency of key mutations, particularly at the AR locus.

Reviewer 1 Major Comment #3 - Subtyping based on mutually-exclusive aberrations

The authors state in lines 155-156 that "This study also showed that a large population of mCRPC patients do not fall into an as-of-yet clinically-relevant or biologically-clear genotype...". One unresolved question is the extent to which the 8 clusters of metastatic CRPC identified in this study correspond to known subtypes of prostate cancer. The authors point out in the Introduction section (lines 51-53) that a similar study of primary prostate cancer (reference #4) was able to classify 74% of cases into 7 defined subtypes based on a small number of mutations. If the authors perform similar subtyping of their 197 metastatic CRPC samples, to what extent do they overlap with each of the known subtypes? It would be a potentially interesting way to gain insight into whether certain types of genomic instability play causal roles in the acquisition of key driver mutations, or vice versa.

Response:

We agree with this insightful remark to better incorporate and highlight the previous mutational clustering of localized prostate cancer as shown by the TCGA7 based on SPOP, FOXA1 and IDH1 mutations and copy-number aberrations and ETS-family gene fusions (and overexpression) per promiscuous partner (ERG, ETV1, ETV4 and FLI1). Half of ETV1, ETV4 and FLI1-clustered samples in the TCGA paper were based on overexpression. Unfortunately, we currently lack matched mRNA-sequencing and therefore cannot observe overexpression of fused ETS-family members, restricting us to only characterize the genomic breaks of these promiscuous partners.

We've included an additional supplementary figure (Supplementary figure 10; lines 217 - 229) showing the presence of these cluster-specific genomic mutations and genomic fusions in our CPCT-02 metastatic mCRPC (WGS) cohort. To reduce differences between the two studies, we only incorporated the mutational categories as used by TCGA and comparable to ours; coding mutations, deep deletions and presence of fusion, ignoring deep gains.

When comparing the frequencies of these cluster-specific aberrations (with and without overexpression status) between the localized prostate cancers from the TCGA paper versus our mCRPC cohort (supplementary figure 10b), we observe that all cluster-specific mutations, except FOXA1 mutations and ETV1 fusions, share similar mutational frequencies. The lower number of ETS-fusions, compared to TCGA, could possibly be attributed to missing matched RNA sequencing which can be more sensitive for fusion-detection, when compared to detecting genomic breaks using WGS, especially for highly-expressed fusions. Interestingly, we do not observe full mutual-exclusivity between ERG-fusion and FOXA1, IDH1, and SPOP mutations nor do we detect IDH1-mutant patients without an accompanying cluster-specific mutation.

From this, we conclude that whole-genome sequencing of the metastatic samples reveals genomic landscapes complementing mutually-exclusive genomic mutations and partially superseding such observed mutations, such as samples without observed pathogenic CDK12 mutation or BRCA2 mutation falling within our CDK12 and BRCA2 cluster (cluster B and D, respectively; lines 249 - 264)

Using this mutation-driven TCGA clustering, we cannot differentiate patients into MSI (WGS cluster A), CDK12 (WGS cluster B) and BRCA2 (WGS cluster D) subgroups as these are spread throughout the TCGA-driven clusters. The resulting ETS-based fusion, SPOP and FOXA1 clusters do not harbor specific genomic landscapes linked to a single species of genomic instability or genotype.

In addition, we could not derive further statistical enrichment or depletion ($q \leq 0.05$; two-sided Fisher's Exact Test with Benjamini-Hochberg corrections) between mutation-driven clusters and additional mutations, kataegis and chromothripsis, except the known enrichment of homozygous CHD1 deletions in the SPOP-cluster (lines 227 - 229).

Reviewer New Comment:

The authors have successfully tested the proposed idea in Supplementary Figure 10. While it did not uncover associations between the novel genomic mutation type-based clusters and the TCGA driver mutation-based clusters, the additional analyses have added to the thoroughness of the study and the ability to integrate its results with previous studies.

Reviewer 1 Major Comment #4 - Re-evaluation of clustering criteria

Many of the insights in the manuscript come from the 4 clusters of metastatic CRPC cases that are associated with a recognized underlying source of genomic instability. On the other hand, the 4 novel clusters undergo an extensive exploration of what they have in common at the genomic level (Supplementary Figures 6-10) but with limited or difficult-to-interpret results. For example, Cluster G is associated with translocations, Cluster H with insertions, Cluster C with tandem duplications <100kb, and Cluster E (the largest cluster) with nothing in particular. Despite the authors' deliberate selection of 8 as the optimal number of clusters (Supplementary Figure 7a), there seems to be a lingering possibility that clustering is not optimally representing the underlying biology. The association of Clusters G and H with translocations and insertions, respectively, reveals how each type of structural variation was given an individual weight as a criterion for clustering, rather than being grouped together as a single marker of chromosomal-level instability, etc. This modeled the assumption that each type of structural variation might have a different underlying cause. However, if structural variations were grouped together, it would be interesting to see if variables like point mutation signatures (which did not associate with any cluster in this

analysis) might have relatively more weight to identify their own cluster. Did the authors try any alternative set of clustering criteria in which two or more types of structural variation were grouped together as a single marker of an over-arching chromosomal instability phenotype? Could they provide some description of the process by which they identified the clustering criteria described in lines 110-111 as the best option?

Response:

The reviewer notes an interesting observation which we investigated further. We performed an additional PCA to identify possible combinations of the proposed clustering metrics (TMB, mean genome ploidy, relative percentage of SV categories and total SV). This PCA (Supplementary figure 6a-c) indeed revealed two principal components (PC 3 and PC 7) with a possible link between SV categories, specifically insertions and translocations, as main contributors of their respective components and as noted by the reviewer.

We combined these two categories (insertions and translocations) into a single metric summarizing the relative frequency of both categories, and removed them as individual features, in an additional clustering analysis using the same methodology (Ward.D method on 1-pearson correlation) (review figure 1).

We then cut this clustering outcome into a maximum of 5 to 15 clusters (optimum according to Elbow's method around 7-9) and tested each of these cut-offs for a correlation or enrichment with a two-sided Fisher's Exact Test (BH) to known onco-genes and tumor suppressors, dN/dS and GISTIC2 enriched genes, kataegis, chromothripsis and ETS fusions. The bottom track of review figure 1 shows the cluster outcomes of the 'original' clustering as shown in the manuscript.

As shown in the bottom tracks, the major subgroups as described in the manuscript (MSI, BRCAness, CDK12 and chromothripsis) were observed without major deviations and only cluster E and F (righthand-side of figure) now show a small mixture and a distinct new cluster of several cluster F samples (chromothripsis) based on the total number of SV. No new clusters (ranging from max. 5 to 15 clusters) could be attributed to another new type of aberration or mutant gene. It would seem that the samples with highest genome instability, most aberrant ploidy, SV and certain combinations of SV are indeed the samples with chromothripsis.

We've also added an additional figure (supplementary figure 6d-j) highlighting, partially, the rationale behind the current criteria for the unsupervised clustering and discussed the potential pitfalls of our clustering analysis (lines 282 - 287).

In this manuscript, we wished to highlight the potential of using genome-wide effects of various genetic drivers such as CDK12 (tandem duplications), DDR-deficiency (high TMB / InDels), BRCA1 (small tandem-duplications), BRCA2 (large deletions) to stratify patients for potential treatments on these effects.

For instance, patients with variants-of-uncertain-significance (VUS) in BRCA2 could potentially benefit from PARP-inhibitors if on genome-wide level the genomic scars (or mutational signatures) associated with pathogenic BRCA2 variants are found.

Future research into identifying clinically-actionable mechanisms will potentially yield new distinct genomic (and mutational) features which can then be incorporated into an overall classifier to better stratify patients. This manuscript shows that this is already currently possible using this approach as a proof-of-concept and shows that this type of stratification will likely only get improved as new features are discovered and more advanced stratification approaches are employed (such as machine learning or 'classical' classifier approaches).

Reviewer New Comment:

The authors have addressed the question well by performing an additional PCA in supplementary

figure 6. The breakdown in supplementary figure 6d-j is also helpful to illustrate the unique criteria defining each cluster.

Reviewer 1 Minor Comment #1 - Y-axis Figure 3c (ChIP-Seq)

Figure 3c would likely be easier to interpret if y-axes for multiple tracks belonging to the same target were scaled to the same maximum values. For example, the untreated and R1881-treated AR ChIP-seq tracks within the AR locus for LNCaP could be shown at the same scale, with the range from 0.0 to 1.2 shown on both y-axes.

Response:

We have scaled the y-axis to the same maximum values for tracks belonging to the same target (AR - LNCaP + AR LNCaP R1881 and AR VCaP + AR VCaP Bicalutamide Resistant per shown loci). Indeed, this provides a better visual distinction between both conditions (Figure 4c).

Reviewer New Comment:

The authors have done some scaling in Figure 4C as described in their response, but have missed some opportunities to do so elsewhere in the figure. I should have written more clearly in the earlier comment that it would also be helpful to consistently scale the FOXA1 tracks (9-11, currently using maximum values ranging from 2.5 to 30 for the AR locus) and H3K27AC tracks (12-14, currently using maximum values ranging from 4 to 60 for the AR locus). It is understandable if the authors choose to scale the LNCaP ChIP-seq tracks differently from the VCaP or patient tracks due to ChIP-seq protocol differences, since it seems that some data have been obtained from a previous manuscript while others were generated specifically for this manuscript. But at least the newly-generated data from Patients A and B should be presented on the same scale for each ChIP target (AR, FOXA1, AND H3K27AC).

Reviewer 2 Major Comment #2 – Treatment history and genetic heterogeneity

Has the treatment history of the patient influenced the observed genetic heterogeneity?

Response:

In this manuscript, the treatment history (number, sequence and type of previous therapies) differs significantly between patients. Therefore, it is not feasible to perform robust statistical analysis on the correlation between genomic data and every individual pre-treatment.

However, to still provide some insight into a possible correlation, we have grouped diverse pre-treatments into ten general treatment groups:

- Only secondary anti-hormonal therapy (with / without radio-nucleotides);
- Only chemo-treatment (with / without radionucleotide);
- Only radionucleotides;
- Secondary anti-hormonal therapy + one chemo-treatment (with / without radio-nucleotides);
- Secondary anti-hormonal therapy + two chemo-treatment (with / without radio-nucleotides);
- No additional treatment after ADT.

After splitting the mCRPC cohort into these 10 groups based on generalized pre-treatment history, we performed a statistical analysis using a two-sided Fisher's Exact Test with Benjamini-Hochberg corrections. We could not detect statistical enrichment or depletion ($q \leq 0.05$) between these supervised treatment groups and our defined genomic clusters (cluster A - H), nor did we observe an enrichment or depletion of mutations, MSI status, kataegis and chromothripsis in relation to treatment group.

We have added these treatment groups as a separate track below an additional figure

reviewFig2_OncoplotWithPretreatment.svg to also visualize that no striking correlation between (general) treatment and genetic heterogeneity can be seen. This could potentially also be in relation to the heterogeneous nature of the current cohort as described in Major Comment #1 from this reviewer.

We've extended the results section with a paragraph on this (lines 150 - 154) and added an additional supplemental table 3 to show the results from this analysis.

Reviewer New Comment:

This is an interesting reviewer question and a satisfactory response by the authors. However, is there any reason not to include the reviewer figure among the supplementary figures for the final manuscript?

Reviewer #2 (Remarks to the Author):

Thank you for taking the time to thoroughly address my previous critiques. I am satisfied with the revised submission

Reviewer #1

Introductory Comment – CHIP-seq data

The study includes other experiments that build on the whole genome sequencing data, particularly ChIP-seq studies that yield some interesting findings such as the validation that certain amplification events observed in patients coincide with genome binding sites for the AR and FOXA1 transcription factors. However, these data are different enough from the title of the manuscript and the theme of demonstrating the clinical utility of whole genome sequencing data that they do not help the study form a single cohesive story. Overall, the study provides a valuable analysis of a large number of metastatic, castration-resistant prostate cancer samples, with subtyping based on somatically acquired genomic changes. Specific comments and suggestions are below.

Response:

Our WGS analysis showed that alterations of the AR-pathway are not subgroup specific. Combining our WGS data with ChIP-seq data shows the potential clinical relevance of recurrent alterations in non-coding regions identified with WGS. The combination of WGS data and ChIP-seq data in patient material, in our view firmly establishes the importance of assessing non-coding regions in prostate cancer. As our manuscript describes the potential clinical impact of WGS based stratification, we feel this analysis does fit within the scope of our manuscript.

Reviewer new comment:

While it is understandable that the authors feel that the ChIP-seq data in Figure 4C are relevant to the study and worth including among the data, it remains difficult to make strong conclusions from them, other than to confirm that some AR, FOXA1, and H3K27ac peaks detectable in cell line models are also present in metastatic patient tissues. We question whether the conclusion that “From WGS and ChIP-seq data, we show that regulators in non-coding regions of MYC and AR are transcriptionally active and highlight the central role of AR signaling in tumor progression” is worth mentioning in the Abstract. On the other hand, the authors’ response makes a good point about “the potential clinical relevance of recurrent alterations in non-coding regions identified with WGS”. Perhaps this is the more Abstract-worthy finding facilitated by the ChIP-seq data? Some shortcomings in the ChIP-seq design of Figure 4C are described below, along with some resulting caveats.

The authors focus on 2 amplifications (at an enhancer near the AR locus and at 8q24.21) observed in the 197-patient WGS dataset and examine those 2 loci by ChIP-seq generated from 2 patients (A and B). Patients A and B were chosen because they exhibited the AR enhancer amplification of interest, but the experiment did not include control tissues that did not exhibit this amplification, either from other metastatic or non-metastatic patients lacking the amplification, or from non-tumor tissue. In fact, the matching peaks in the stimulated LNCaP (without AR amplification) and VCaP cells (with AR amplification) indicate that the presence of AR, FOXA1, or H3K27 signal at this enhancer is not unique to prostate cancer cells with AR enhancer amplification. The reader can only conclude that the amplification event coincides with an enhancer region that is marked by AR, FOXA1, or H3K27ac signal. The reader’s confidence that the amplification is driving the amount of AR, FOXA1, or H3K27 signal is highly dependent on existing literature about AR enhancer amplifications. The authors have generated ChIP-seq data from Patients A and B that are consistent with the literature and with the cell line controls, but are not well-controlled enough or sufficiently complemented by RNA-level data to enable stand-alone conclusions. The discussion of Figure 4C should be expanded to mention some of these caveats.

The presentation of results from 8q24.21 (also in Figure 4C) is slightly problematic too, in the sense that the Abstract mentions MYC. One of the challenges of interpreting ChIP-seq data is the difficulty of confidently assigning signal to a particular gene based on proximity alone, unless complementary RNA level data is presented showing that the gene expression is dependent on the presence/absence of the ChIP target.

Without more direct evidence that PCAT1 or MYC expression is dependent on the presence of AR or FOXA1, it would not be possible to confidently conclude that the observed peaks control the expression of either gene. One way to strengthen this could be to compare the proposed MYC peak location with at least one other study reporting that AR binds to the MYC locus to control its expression (Gao L, Schwartzman J, Gibbs A, Lisac R, Kleinschmidt R, Wilmot B, Bottomly D, Coleman I, Nelson P, McWeeney S, Alumkal J. Androgen receptor promotes ligand-independent prostate cancer progression through c-Myc upregulation. PLoS One. 2013 May 21;8(5):e63563.). Possibly the authors could determine whether the amplified region coincides with a MYC-regulating AR binding site characterized in that study? This would strengthen the Abstract's currently too-strong attribution of this amplified region to the control of the MYC locus.

Response:

We agree with the reviewer that we currently do not show enough direct evidence that the amplified region near *PCAT1/MYC*, with co-occupancy of AR and FOXA1, indeed affects *MYC* expression. We have altered the abstract, results and discussion to highlight this shortcoming in using only WGS and CHIP-Seq data without further experimental data.

We also added a reference to a recent study by Mazrooei, P. *et al.* (Accepted in Cancer Cell 2019)¹ in which the authors show that the observed *PCAT1/MYC* region and the *MYC* locus are part of a topologically associated domain (figure 3a of their manuscript, termed CRE2 in their study). Subsequently, they elegantly show with CRISPR-Cas9 deletions of CRE2, followed by measuring the *MYC* expression with luciferase assays, that this region has an effect on *MYC* expression in VCaP prostate cancer cells (figure 3e-f of their manuscript). Deleting the CRE2 shows down-regulation of *MYC*.

Again, this does not provide direct evidence of the role of the amplification of this same region in regulating *MYC* expression in mCRPC-patient settings, but does further hint towards a plausible functional effect on *MYC*.

As the reviewer suggested adding several non-mCRPC patients (or mCRPC patients lacking amplifications), we have added three independent hormone-sensitive primary adenocarcinoma (PRAD) samples which underwent the same CHIP-seq protocols regarding AR, FOXA1 and H3K27ac (Supplementary figure 7).

We show two known AR-regulated positive controls (*TMPRSS2* and *KLK3*) and the two mCRPC-derived amplified regions; *AR* and *PCAT1/MYC* as denoted with a red asterisk. Within these regions, we observe reduced levels of AR, FOXA1 and H3K27ac co-occupancy in all three PRAD patients which could indicate that these are not used as putative enhancers in primary settings. With that, our study confirms previous findings that the AR cistrome deviates between primary prostate cancers and progressive disease by Stelloo *et al.*, 2015². Furthermore, as the AR enhancer is gained in copy number which is causally linked with increased enhancer signaling and thus higher AR mRNA expression levels, this mCRPC-specific enhancer de-regulation may be a wider-spread -phenomenon not observed in the primary disease setting.

Changes in the manuscript (Final Markup) regarding this issue:

- r52-54 (Abstract):
 - Took over the suggestion of the reviewer about rephrasing the *AR/MYC* statement as “the potential clinical relevance of recurrent alterations in non-coding regions identified with WGS”
- r159-181 (Results - The role of the AR-pathway in mCRPC):
 - Incorporated the CHIP-seq results of the three independent PRAD samples showing reduced levels of co-occupancy.
- r272-r284 (Discussion)
 - Added the citation to the Mazrooei, P. *et al.* study on the *MYC* region.
 - Added the drawback:

- “However, the presented WGS and CHIP-seq data are not conclusive in elucidating the definitive role of this amplified region in regulating *MYC* expression and further experiments are needed to establish a potential link to *MYC* regulation.”
- Added Supplementary figure 7 (three independent PRAD CHIP-seq samples showing *TMPRSS2*, *KLK3*, *AR* and *MYC* regions with AR, FOXA1 and H3K27ac occupancies)

Minor Comment #1 - Y-axis Figure 4c (ChIP-Seq)

Figure 3c would likely be easier to interpret if y-axes for multiple tracks belonging to the same target were scaled to the same maximum values. For example, the untreated and R1881-treated AR ChIP-seq tracks within the AR locus for LNCaP could be shown at the same scale, with the range from 0.0 to 1.2 shown on both y-axes.

Response:

We have scaled the y-axis to the same maximum values for tracks belonging to the same target (AR - LNCaP + AR LNCaP R1881 and AR VCaP + AR VCaP Bicalutamide Resistant per shown loci). Indeed, this provides a better visual distinction between both conditions (Figure 4c).

Reviewer new comment:

The authors have done some scaling in Figure 4C as described in their response, but have missed some opportunities to do so elsewhere in the figure. I should have written more clearly in the earlier comment that it would also be helpful to consistently scale the FOXA1 tracks (9-11, currently using maximum values ranging from 2.5 to 30 for the AR locus) and H3K27AC tracks (12-14, currently using maximum values ranging from 4 to 60 for the AR locus). It is understandable if the authors choose to scale the LNCaP ChIP-seq tracks differently from the VCaP or patient tracks due to ChIP-seq protocol differences, since it seems that some data have been obtained from a previous manuscript while others were generated specifically for this manuscript. But at least the newly-generated data from Patients A and B should be presented on the same scale for each ChIP target (AR, FOXA1, AND H3K27AC).

Response:

We have adjusted the scales per each ChIP-target for patients A and B as suggested by the reviewer, for both the AR and MYC regions in figure 4c.

Reviewer #2

Major Comment #1 - Correlation to survival

Is there any correlation between the identified genotypes and overall survival?

Response:

The CPCT-02 study on which we are reporting collected biopsies of metastatic prostate cancer patients irrespective of the timing with respect to their stage of the disease. This has led to a broad heterogeneity of number and type of prior treatments (see response to Major comment #2 of this reviewer). Therefore, the prognosis of patients differs greatly depending on the treatment history, as the treatments for mCRPC patients nowadays significantly impacts overall survival.

However, one may argue that the genomic data for each patient is relatively stable and therefore survival analysis based on the classification into subgroups may still be reasonable. In our opinion there is ample evidence that the genomic data derived from patient alters over time and some well-defined prognostic factors such as p53 mutations, Rb1 loss and AR aberrations will change over time. Therefore, we do not believe that reporting survival data in the current manuscript allows for an improved understanding without correcting for all possible biases or setting up a more homogeneous patient cohort. These biases will be comprised of genomic features that have changed over time (as illustrated in response to 'Major Comment #1' from Reviewer #1). The subgroup determination may represent a stable factor; however, this is subject to future studies in which we will try to obtain biopsies from the same patient at multiple time points to prove the subgroup classification is stable over the course of the disease. Moreover, the clinical features that potentially affect prognosis such as whether the patient had initially high-risk metastatic disease or had progressed years after local treatment would need to be incorporated and is currently not (yet) available for this cohort. Treatment response factors such as the time on treatment for the first ADT treatment, disease load as measured by alkaline phosphatase, lactate hydrogenase levels and other general prognostic factors or response to later lines of therapy may also affect prognosis. The description of all of these clinical features is beyond the scope of this manuscript and therefore we feel reporting survival data without the proper context will not add value to this manuscript and would represent an incomplete view on all these varying factors.

We did incorporate a section in the discussion to reflect that the lack of homogeneity of the population makes analyses on the prognostic value of these data inappropriate (lines 296 - 302).

Reviewer new comment:

The original Reviewer 1 Comment 1 was proposed as a way to gain insight into clinical outcomes associated with each of the 8 subtypes identified by the authors, through estimation of the proportion of each subtype found in earlier-stage tumors, and determination of their relative enrichment or depletion from the metastatic tumors. From the authors' response to that comment and to the related Reviewer 2 Comment 2, it seems that this was performed satisfactorily but that the reviewer's underlying question was misguided or impossible to evaluate in this manner, in that these subtypes were either present in the primary prostate cancer dataset at much reduced frequencies and/or were difficult to detect (no samples with CDK12 coding mutations, few with MSI or coding mutations in BRCA2) due to low TMB, etc. Perhaps a non-metastatic CRPC cohort would be more appropriate for the proposed comparison. But this would be outside the scope of the present study, and the additional analyses are satisfactory. Nevertheless, as Reviewer 2 Comment 1 also points out, the inability to link any of the 8 clusters/subtypes from this study to clinical outcome relative to the others remains at least a minor shortcoming. The literature suggests that the type of genomic instability is potentially clinically meaningful, at least as a predictor of a sustained response to immune checkpoint blockade. However, in the present study, the comparison to primary tumors instead has indicated simply that tumor mutation burden, regardless of origin, is strongly associated with progression to metastatic CRPC.

Response:

We agree with both reviewers that the next insights would be gained by correlating the findings from this manuscript, and others, to overall patient survival or clinical attributes as a future study. We are currently investigating this as future work but, in accordance with the reviewers, also deem this as out-of-scope for this current manuscript.

Major Comment #2 – Treatment history and genetic heterogeneity

Has the treatment history of the patient influenced the observed genetic heterogeneity?

Response:

In this manuscript, the treatment history (number, sequence and type of previous therapies) differs significantly between patients. Therefore, it is not feasible to perform robust statistical analysis on the correlation between genomic data and every individual pre-treatment.

However, to still provide some insight into a possible correlation, we have grouped diverse pre-treatments into ten general treatment groups:

- Only secondary anti-hormonal therapy (with / without radio-nucleotides);
- Only chemo-treatment (with / without radionucleotide);
- Only radionucleotides;
- Secondary anti-hormonal therapy + one chemo-treatment (with / without radio-nucleotides);
- Secondary anti-hormonal therapy + two chemo-treatment (with / without radio-nucleotides);
- No additional treatment after ADT.

After splitting the mCRPC cohort into these 10 groups based on generalized pre-treatment history, we performed a statistical analysis using a two-sided Fisher's Exact Test with Benjamini-Hochberg corrections. We could not detect statistical enrichment or depletion ($q \leq 0.05$) between these supervised treatment groups and our defined genomic clusters (cluster A - H), nor did we observe an enrichment or depletion of mutations, MSI status, kataegis and chromothripsis in relation to treatment group.

We have added these treatment groups as a separate track below an additional figure `reviewFig2_OncoplotWithPretreatment.svg` to also visualize that no striking correlation between (general) treatment and genetic heterogeneity can be seen. This could potentially also be in relation to the heterogeneous nature of the current cohort as described in Major Comment #1 from this reviewer.

We've extended the results section with a paragraph on this (lines 150 - 154) and added an additional supplemental table 3 to show the results from this analysis.

Reviewer new comment:

This is an interesting reviewer question and a satisfactory response by the authors. However, is there any reason not to include the reviewer figure among the supplementary figures for the final manuscript?

Response:

After reconsideration, we have indeed included this figure as Supplemental figure 6 within this manuscript as it does indeed address an interesting observation on patient heterogeneity.

References

1. Mazrooei, P. *et al.* Somatic Mutations and Risk-Variants Converge on Cis-Regulatory Elements to Reveal the Cancer Driver Transcription Regulators in Primary Prostate Tumors. *SSRN Electron. J.* (2018). doi:10.2139/ssrn.3245213
2. Stelloo, S. *et al.* Androgen receptor profiling predicts prostate cancer outcome. *EMBO Mol. Med.* (2015). doi:10.15252/emmm.201505424